# Task-Aware Preference Calibration for Direct Preference Optimization

**Mengyang Li**[1]   **Zhong Zhang**[1]   **Pinlong Zhao**[2]

## Abstract

Direct Preference Optimization (DPO) has become a predominant approach for aligning large language models with human preferences. Recent work has used perplexity differentials to identify unreliable preference labels, but these methods apply uniform calibration strategies across all samples. We observe that the reliability of perplexity signals varies substantially across task types: perplexity differentials strongly correlate with preference quality for factual tasks but provide weak signals for creative tasks where novelty is valued. Based on this observation, we propose Task-Aware Preference Calibration (TAPC), which learns task-conditioned calibration functions that adapt to the characteristics of different prompt types. TAPC employs a task encoder to extract prompt representations and learns task-specific slope and bias parameters for mapping perplexity signals to confidence targets. Through meta-learning on a small reference dataset, TAPC discovers how to weight perplexity signals appropriately for each task category. Experiments on Llama-3-8B and Qwen2-7B demonstrate that TAPC outperforms existing methods across multiple benchmarks, with particularly large improvements on creative and open-ended tasks where uniform calibration strategies fail.

## 1. Introduction

Aligning large language models (LLMs) with human preferences has emerged as a critical challenge in developing reliable AI systems (Bai et al., 2022; Ouyang et al., 2022). Direct Preference Optimization (DPO) (Rafailov et al., 2023) provides an efficient alternative to reinforcement learning

[1]Tianjin Key Laboratory of Wireless Mobile Communications and Power Transmission, Tianjin Normal University, Tianjin, China [2]School of Cyberspace, Hangzhou Dianzi University, Hangzhou, China. Correspondence to: Pinlong Zhao <pinlongzhao@hdu.edu.cn>.

*Proceedings of the 43rd International Conference on Machine Learning*, Seoul, South Korea. PMLR 306, 2026. Copyright 2026 by the author(s).

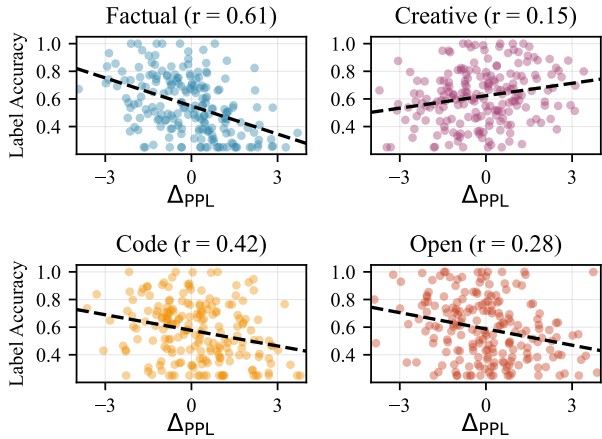

*Figure 1.* Reliability of perplexity differential ($\Delta_{\text{PPL}}$) as a label quality signal varies across task types. Factual tasks show strong negative correlation with label correctness ($r = -0.61$, $p < 0.001$), while creative tasks show weak correlation ($r = -0.15$, $p < 0.001$ but negligible effect size). Subplot titles report $|r|$ for readability.

from human feedback (RLHF), directly optimizing policies using pairwise preference data without explicit reward modeling. The simplicity and effectiveness of DPO have inspired numerous extensions (Ethayarajh et al., 2024; Azar et al., 2024; Meng et al., 2024; Song et al., 2024).

A recognized limitation of DPO is its assumption that all preference labels are equally reliable. Human preference data exhibits substantial heterogeneity, where some pairs represent clear consensus while others capture marginal distinctions or genuine annotator disagreement (Bai et al., 2022; Köpf et al., 2023; Gao et al., 2024). Recent work has proposed using perplexity differentials to identify potentially mislabeled examples (Li et al., 2024). The intuition is that well-trained models assign lower perplexity to genuinely better responses, so disagreement between perplexity signals and preference labels may indicate annotation errors.

However, existing approaches apply uniform calibration strategies across all samples, implicitly assuming that perplexity differentials are equally informative regardless of task type. We argue that this assumption is fundamentally flawed. Consider the difference between factual question answering and creative writing. For factual tasks, correct answers tend to appear frequently in pretraining corpora,

making perplexity a reliable quality indicator. For creative tasks, high-quality responses may exhibit novelty and originality that actually correlates with higher perplexity. A poem that skillfully subverts expectations or a story with an unexpected twist may be genuinely preferred despite receiving higher perplexity scores.

To validate this observation, we conduct a preliminary analysis on a subset of UltraFeedback where we obtain high-confidence ground-truth labels through majority voting among multiple evaluators (see Appendix A for details). Figure 1 shows that the correlation between perplexity differentials and label correctness varies dramatically across task types. Factual tasks exhibit strong negative correlation (Pearson $r = -0.61$, $p < 0.001$), indicating that lower perplexity reliably signals correct labels, while creative tasks show only weak correlation ($r = -0.15$, $p < 0.001$; statistically significant due to large $n$ but with negligible effect size). This finding suggests that effective preference calibration must account for task-specific characteristics.

Based on this insight, we propose Task-Aware Preference Calibration (TAPC). Instead of learning a single mapping from perplexity signals to confidence targets, TAPC learns task-conditioned calibration functions. Our approach consists of three components. First, a task encoder extracts representations from prompts that capture task-relevant characteristics. Second, a task-conditioned calibration module predicts instance-specific confidence targets using both perplexity signals and task representations. Third, a meta-learning procedure trains the calibration module to produce targets that improve policy performance on trusted reference examples.

The key technical innovation is the parameterization of task-conditioned calibration. We learn task-specific slope and bias parameters that modulate how perplexity differentials map to confidence scores, providing interpretability: we can examine how different task types weight perplexity signals. Theoretical analysis establishes that task-aware approaches achieve lower calibration error than task-agnostic methods when optimal calibration varies across tasks. Experiments on Llama-3-8B and Qwen2-7B demonstrate that TAPC outperforms DPO and recent alternatives across multiple benchmarks, with the largest improvements on creative and open-ended tasks where uniform perplexity-based calibration fails. Analysis reveals interpretable patterns: TAPC learns steep slopes for factual tasks (trusting perplexity) and flat slopes for creative tasks (discounting perplexity).

Our contributions are summarized as follows:

- We identify that perplexity-based confidence signals have task-dependent reliability: they strongly indicate label quality for factual tasks but provide weak signals for creative tasks.

- We propose Task-Aware Preference Calibration (TAPC), a framework that learns task-conditioned calibration functions through meta-learning. TAPC uses interpretable task-specific slope and bias parameters that reveal how different tasks should weight perplexity signals.

- We provide theoretical analysis establishing when task-aware calibration outperforms task-agnostic approaches, and demonstrate consistent empirical improvements across models and benchmarks, with particularly large gains on creative tasks.

**Conflict of Interest Disclosure.** The authors declare no financial conflicts of interest related to this work.

## 2. Related Work

**Preference Optimization for LLMs.** Learning from human preferences has evolved significantly since early work on reward learning (Christiano et al., 2017). Traditional RLHF pipelines train reward models on preference data followed by policy optimization (Ouyang et al., 2022; Stiennon et al., 2020). DPO (Rafailov et al., 2023) simplified this pipeline by deriving a closed-form relationship between optimal policies and preferences. Subsequent work proposed various modifications: IPO (Azar et al., 2024) avoids Bradley-Terry assumptions, KTO (Ethayarajh et al., 2024) learns from binary signals, SimPO (Meng et al., 2024) uses reference-free formulations, and ORPO (Hong et al., 2024) combines supervised fine-tuning with preference optimization. More recently, PRO (Song et al., 2024) extends preference optimization to ranking settings with multiple responses, and Li et al. (2026) propose meta-learning paradigms for fusing intrinsic feedback during alignment. Recent surveys provide comprehensive overviews of this rapidly evolving field. Our work addresses an orthogonal challenge: adapting calibration strategies to task characteristics.

**Robust Preference Learning.** Recognition that preference data contains noise has motivated robust learning approaches. Mitchell (2023) proposed cDPO with label smoothing controlled by estimated noise rates. Furuta et al. (2024) developed rDPO with modified loss functions for outlier tolerance. Wu et al. (2024) explored beta distribution parameterizations for modeling soft labels. Li et al. (2024) introduced PerpCorrect, using perplexity differentials to identify and correct mislabeled examples before training. Recent advances include Dr. DPO (Wu et al., 2025a), which applies distributional robustness to handle preference noise, and LPC (Gong et al., 2025), which models latent preference structures to capture annotator heterogeneity. Theoretical analysis of noise robustness has also advanced (Chowdhury

et al., 2024), along with practical methods for detecting low-quality preferences (Kong et al., 2024). While these methods improve robustness through various mechanisms, they apply uniform strategies across all samples. Our work reveals that calibration reliability is task-dependent and proposes methods that adapt accordingly.

**Task-Aware Learning.** The observation that different tasks require different learning strategies has a long history in machine learning. Multi-task learning (Caruana, 1997) shares representations across related tasks while allowing task-specific components. Meta-learning approaches learn to adapt quickly to new tasks (Finn et al., 2017). In the context of language models, task-specific adapters (Houlsby et al., 2019) and prompt-based methods (Lester et al., 2021) enable efficient task adaptation. Self-play methods (Wu et al., 2025b) and iterative refinement approaches have also been explored for preference learning. Our work applies task-awareness to preference calibration, learning how to weight uncertainty signals differently for different task types.

**Meta-Learning for Noisy Labels.** Meta-learning has proven effective for handling noisy supervision in classification. Meta-Weight-Net (Shu et al., 2019) learns instance weights through meta-gradient descent. MentorNet (Jiang et al., 2018) trains curriculum networks to select clean examples. More broadly, understanding training dynamics (Li et al., 2025) and predicting learning trajectories (Li & Zhao, 2026) provide complementary perspectives on how models interact with data of varying quality. These methods typically assume task-homogeneous noise. Our work extends meta-learning to preference optimization with task-heterogeneous uncertainty, where different task types exhibit different noise characteristics and require different calibration strategies.

## 3. Preliminaries

### 3.1. Direct Preference Optimization

Direct Preference Optimization (Rafailov et al., 2023) provides an efficient approach to aligning language models with human preferences. Given a dataset $\mathcal{D} = \{(x^{(i)}, y_w^{(i)}, y_l^{(i)})\}_{i=1}^N$ where $x$ denotes a prompt, $y_w$ denotes the preferred response, and $y_l$ denotes the dispreferred response, DPO optimizes the policy $\pi_\theta$ by minimizing:

$$\mathcal{L}_{\text{DPO}} = -\mathbb{E}_{(x,y_w,y_l)\sim\mathcal{D}} \left[ \log \sigma(\beta h_\theta) \right], \quad (1)$$

where $\sigma(\cdot)$ is the sigmoid function, $\beta$ is a temperature parameter, and $h_\theta$ is the log-ratio defined as:

$$h_\theta(x, y_w, y_l) = \log \frac{\pi_\theta(y_w|x)}{\pi_{\text{ref}}(y_w|x)} - \log \frac{\pi_\theta(y_l|x)}{\pi_{\text{ref}}(y_l|x)}. \quad (2)$$

Here $\pi_{\text{ref}}$ is a reference policy, typically the supervised fine-tuned model. The DPO objective corresponds to maximum likelihood estimation under the Bradley-Terry preference model (Bradley & Terry, 1952).

### 3.2. Soft-Label Formulation

The standard DPO objective implicitly assigns a target probability of 1 to each observed preference. This can be generalized using cross-entropy with soft targets. Let $P_\theta = \sigma(\beta h_\theta)$ denote the model's implied preference probability. For a target confidence $\hat{p} \in [0, 1]$, the cross-entropy loss is:

$$\mathcal{L}_{\text{CE}} = -\mathbb{E} \left[ \hat{p} \log P_\theta + (1 - \hat{p}) \log(1 - P_\theta) \right]. \quad (3)$$

When $\hat{p} = 1$, this reduces to standard DPO. Setting $\hat{p} < 1$ implements label smoothing, reducing confidence in potentially unreliable labels.

### 3.3. Perplexity Differential as Uncertainty Signal

Prior work has identified perplexity differentials as informative signals for preference reliability (Li et al., 2024). Given an anchor model $\pi_{\text{anchor}}$ (typically the SFT model), the perplexity differential for a preference pair is:

$$\Delta_{\text{PPL}} = \log \text{PPL}(y_w; \pi_{\text{anchor}}) - \log \text{PPL}(y_l; \pi_{\text{anchor}}), \quad (4)$$

where $\text{PPL}(y; \pi)$ is the perplexity of response $y$ under model $\pi$. Negative values of $\Delta_{\text{PPL}}$ indicate that the preferred response has lower perplexity, consistent with the label. Positive values suggest potential label inconsistency.

The critical observation motivating our work is that the reliability of $\Delta_{\text{PPL}}$ as a confidence signal varies across task types. We formalize and address this heterogeneity in the following sections.

## 4. Task-Aware Preference Calibration

We present Task-Aware Preference Calibration (TAPC), our framework for learning task-conditioned calibration functions. We first analyze the task-dependence of perplexity signals, then describe the TAPC architecture and meta-learning procedure, and finally provide theoretical justification.

### 4.1. Task-Dependent Reliability of Perplexity Signals

We begin by formalizing the observation that perplexity differentials have varying reliability across task types. Let $\tau : \mathcal{X} \to \mathcal{T}$ denote a mapping from prompts to task categories, and let $c^*(x, y_w, y_l) \in [0.5, 1]$ denote the true probability that the observed preference label is correct.

**Definition 4.1** (Task-Conditional Calibration Function). For a task category $t \in \mathcal{T}$, the optimal calibration function $f_t^* : \mathbb{R} \to [0, 1]$ maps perplexity differentials to confidence

targets:

$$f_t^*(\Delta_{\text{PPL}}) = \mathbb{E}\left[c^*(x, y_w, y_l) \mid t(x) = t, \Delta_{\text{PPL}}\right], \quad (5)$$

where $t(x)$ denotes the task category of prompt $x$.

The following proposition establishes that optimal calibration functions differ across tasks.

**Proposition 4.2** (Task Heterogeneity of Optimal Calibration). *Let $\mathcal{T} = \{t_{fact}, t_{creative}\}$ denote factual and creative task categories. Under the following conditions:*

1. *For factual tasks, correct responses have systematically lower perplexity: $\mathbb{E}[\Delta_{PPL} \mid label\ correct, t_{fact}] < 0$.*

2. *For creative tasks, high-quality responses may have higher perplexity due to novelty: $\mathbb{E}[\Delta_{PPL} \mid label\ correct, t_{creative}] \approx 0$.*

*Then the optimal calibration functions satisfy $f_{t_{fact}}^* \neq f_{t_{creative}}^*$.*

This result implies that any task-agnostic calibration function $f$ that ignores task information incurs suboptimal calibration error on at least one task category. The proof is provided in Appendix C.1.

### 4.2. TAPC Architecture

Figure 2 illustrates the TAPC framework. The system consists of three components: a task encoder that extracts prompt representations, a task-conditioned calibration module that predicts confidence targets, and the policy LLM being aligned.

**Task Encoder.** The task encoder $E_\psi : \mathcal{X} \to \mathbb{R}^d$ maps prompts to task representations. We implement this using the hidden states from the reference LLM:

$$\mathbf{z}(x) = \text{MLP}_\psi \left( \frac{1}{|x|} \sum_{i=1}^{|x|} \mathbf{h}_i^{(\text{ref})} \right), \quad (6)$$

where $\mathbf{h}_i^{(\text{ref})}$ denotes the hidden state at position $i$ from the reference model, and $\text{MLP}_\psi$ is a learnable projection to a $d$-dimensional task space. We use $\mathbf{z}(x) \in \mathbb{R}^d$ to denote the learned task embedding, distinguishing it from the discrete task category $t(x) \in \mathcal{T}$.

**Task-Conditioned Calibration.** The key innovation of TAPC is learning task-specific calibration parameters. Rather than using a generic network, we parameterize the calibration function as:

$$\hat{p} = \sigma\left(\gamma(\mathbf{z}(x)) \cdot \Delta_{\text{PPL}} + \mu(\mathbf{z}(x))\right), \quad (7)$$

where $\gamma : \mathbb{R}^d \to \mathbb{R}$ and $\mu : \mathbb{R}^d \to \mathbb{R}$ are learned functions that output task-specific slope and bias:

$$\gamma(\mathbf{z}) = \mathbf{w}_\gamma^\top \mathbf{z} + b_\gamma, \quad (8)$$
$$\mu(\mathbf{z}) = \mathbf{w}_\mu^\top \mathbf{z} + b_\mu. \quad (9)$$

This parameterization provides interpretability. The slope $\gamma(\mathbf{z}(x))$ controls how strongly perplexity differentials influence confidence: large positive slopes indicate high trust in perplexity signals, while slopes near zero indicate that perplexity should be discounted. The bias $\mu(\mathbf{z}(x))$ captures task-specific baseline confidence levels. Importantly, because $\gamma$ and $\mu$ are continuous functions of the learned embedding $\mathbf{z}(x)$, TAPC produces instance-specific calibration parameters for each individual sample rather than applying uniform parameters within discrete task categories.

We denote the combined calibration module parameters as $\phi = \{\psi, \mathbf{w}_\gamma, b_\gamma, \mathbf{w}_\mu, b_\mu\}$.

### 4.3. Policy Optimization with Calibrated Targets

Given confidence targets, the policy is optimized using cross-entropy:

$$\mathcal{L}_{\text{policy}} = -\mathbb{E}_{n \sim \mathcal{B}} \left[ \hat{p}_n \log P_\theta^{(n)} + (1 - \hat{p}_n) \log(1 - P_\theta^{(n)}) \right], \quad (10)$$

where $P_\theta^{(n)} = \sigma(\beta h_\theta^{(n)})$. This formulation naturally handles the spectrum of confidence levels. When $\hat{p}_n \approx 1$ (high confidence, typical for clear factual preferences), the loss strongly encourages the observed preference. When $\hat{p}_n \approx 0.5$ (low confidence, typical for ambiguous creative comparisons), the gradient is attenuated. When $\hat{p}_n < 0.5$ (suggesting likely mislabeling), the loss encourages the opposite preference.

### 4.4. Meta-Learning the Calibration Module

The calibration module parameters $\phi$ are learned through meta-optimization. The key idea is to train the module such that the confidence targets it produces lead to good policy performance on a trusted reference dataset $\mathcal{D}_{\text{meta}}$. At each training iteration $t$, the procedure operates in three phases.

**Phase 1: Virtual Policy Update.** Using current calibration parameters $\phi^{(t)}$, we compute confidence targets for a training batch $\mathcal{B}_{\text{train}}$ and perform a virtual gradient step:

$$\theta_{\text{virt}}^{(t+1)} = \theta^{(t)} - \alpha \nabla_\theta \mathcal{L}_{\text{policy}}(\theta^{(t)}, \phi^{(t)}), \quad (11)$$

where $\alpha$ is the policy learning rate. The virtual parameters $\theta_{\text{virt}}^{(t+1)}$ depend on $\phi^{(t)}$ through the confidence targets.

**Phase 2: Meta-Gradient Update.** We evaluate the virtual policy on a batch $\mathcal{B}_{\text{meta}}$ from the reference dataset using

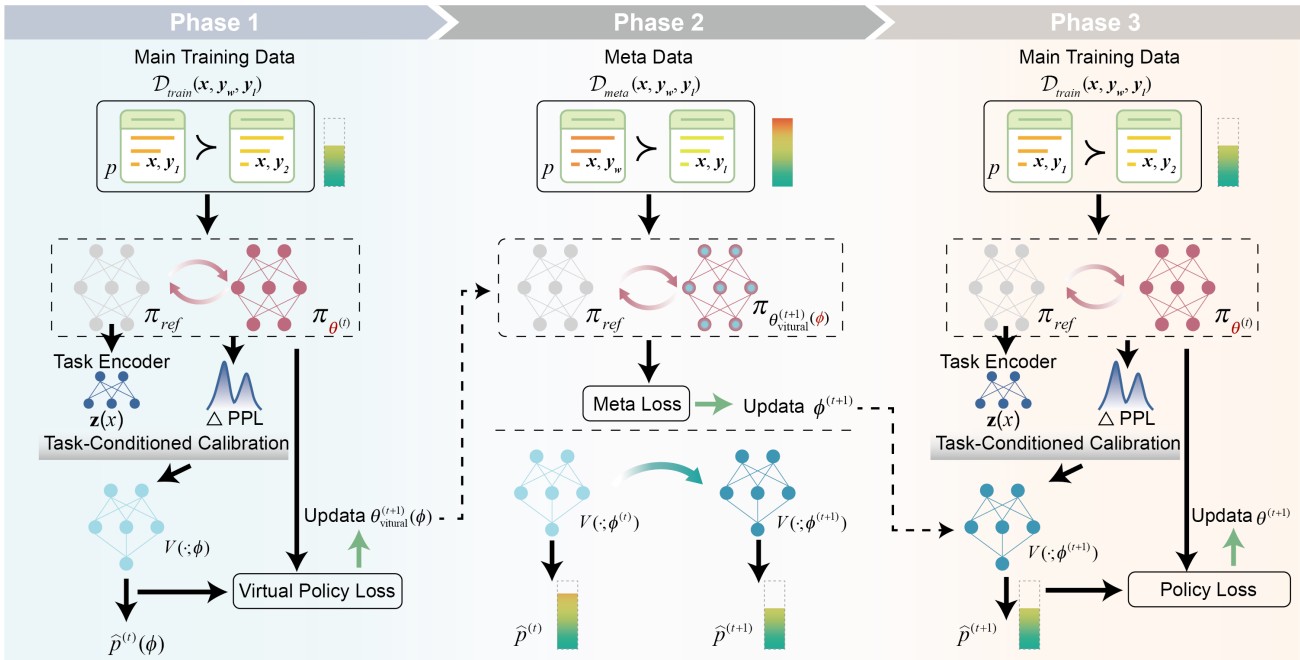

*Figure 2.* Overview of Task-Aware Preference Calibration (TAPC). The task encoder extracts prompt representations $\mathbf{z}(x)$, which condition the calibration module to produce task-specific slope $\gamma$ and bias $\mu$. These parameters modulate how perplexity differentials are transformed into confidence targets $\hat{p}$.

standard DPO loss (assuming clean labels):

$$\mathcal{L}_{\text{meta}}(\phi^{(t)}) = \mathcal{L}_{\text{DPO}}(\theta_{\text{virt}}^{(t+1)}(\phi^{(t)}); \mathcal{B}_{\text{meta}}). \quad (12)$$

The calibration module is updated by descending the meta-gradient:

$$\phi^{(t+1)} = \phi^{(t)} - \eta\nabla_\phi\mathcal{L}_{\text{meta}}(\phi^{(t)}), \quad (13)$$

where $\eta$ is the meta-learning rate. This gradient flows through the virtual update, training the calibration module to assign confidence targets that improve reference set performance.

**Phase 3: Actual Policy Update.** With updated calibration parameters, we recompute confidence targets and update the policy:

$$\theta^{(t+1)} = \theta^{(t)} - \alpha\nabla_\theta\mathcal{L}_{\text{policy}}(\theta^{(t)}, \phi^{(t+1)}). \quad (14)$$

Through this procedure, the calibration module learns task-appropriate calibration strategies. For factual tasks where perplexity signals are reliable, it learns large slopes $\gamma$. For creative tasks where perplexity signals are unreliable, it learns slopes near zero, effectively ignoring the perplexity signal and defaulting to moderate confidence.

### 4.5. Theoretical Analysis

We provide theoretical justification for task-aware calibration, establishing conditions under which it outperforms task-agnostic approaches.

**Theorem 4.3** (Benefit of Task-Aware Calibration). *Let $\mathcal{T} = \{t_1, \ldots, t_K\}$ be a partition of prompts into $K$ task categories with mixing weights $\{p_k\}_{k=1}^K$. Let $f_k^*$ denote the optimal calibration function for task $k$, and let $f^*$ denote the optimal task-agnostic calibration function. Define the calibration error as $\mathcal{E}(f) = \mathbb{E}[(f(\Delta_{PPL}) - c^*)^2]$.*

*If the optimal task-specific functions are heterogeneous, i.e., $\exists k, k'$ such that $f_k^* \neq f_{k'}^*$, then:*

$$\sum_{k=1}^K p_k\mathcal{E}(f_k^*) < \mathcal{E}(f^*). \quad (15)$$

The proof follows from Jensen's inequality applied to the squared error decomposition. When task-specific optima differ, any single function cannot simultaneously minimize error across all tasks.

**Corollary 4.4** (Characterization of Improvement). *The improvement from task-aware calibration is proportional to the between-task variance of optimal calibration functions:*

$$\mathcal{E}(f^*) - \sum_{k=1}^K p_k\mathcal{E}(f_k^*) = \sum_{k=1}^K p_k\mathbb{E}\left[(f_k^*(\Delta_{PPL}) - f^*(\Delta_{PPL}))^2\right].$$
$$(16)$$

This characterization provides insight into when task-awareness helps most: the benefit is largest when different tasks require substantially different calibration strategies.

Concretely, if we denote the slope difference between task types as $\Delta\gamma = |\gamma_{t_1}^* - \gamma_{t_2}^*|$, the improvement scales approximately as $O(\Delta\gamma^2 \cdot \text{Var}(\Delta_{\text{PPL}}))$. In our experiments, Factual and Creative tasks exhibit $\Delta\gamma \approx 1.9$ (Table in Appendix F.2), explaining the large gains observed on these contrasting task types.

We also establish sample complexity for learning task-conditioned calibrators.

**Theorem 4.5** (Sample Complexity). *Let the task encoder output dimension be $d$ and the calibration module have $p$ parameters. With $m$ reference samples and standard regularity conditions, the learned calibrator $\hat{f}$ satisfies with probability at least $1 - \delta$:*

$$\mathbb{E}[\mathcal{E}(\hat{f})] - \min_{f \in \mathcal{F}} \mathbb{E}[\mathcal{E}(f)] \leq O\left(\sqrt{\frac{(d + p)\log m + \log(1/\delta)}{m}}\right). \tag{17}$$

The bound shows that learning task-conditioned calibration requires only modestly more samples than task-agnostic calibration, with the additional cost scaling with the task embedding dimension $d$.

# 5. Experiments

We conduct experiments to evaluate TAPC, focusing on three questions: (1) Does TAPC improve alignment quality over existing methods? (2) Does task-aware calibration provide benefits over task-agnostic approaches? (3) What calibration patterns does TAPC learn for different task types?

## 5.1. Experimental Setup

**Datasets.** We evaluate on two preference datasets with diverse task types. The UltraFeedback dataset (Cui et al., 2024) contains approximately 64K preference pairs spanning multiple task categories including factual QA, creative writing, coding, and reasoning. We use the official splits with 61K training and 3K test pairs. The Golden HH dataset (Bai et al., 2022) contains 116K training pairs from the Anthropic helpful-harmless collection. Following prior work (Li et al., 2024), we evaluate robustness under synthetic label noise at rates $\epsilon \in \{0\%, 10\%, 20\%, 30\%, 40\%\}$.

For task-stratified analysis, we categorize prompts into four types based on keywords and formatting: Factual (questions with objective answers, e.g., containing "what is", "how many", "when did"), Creative (writing, storytelling, poetry, containing "write", "create", "imagine"), Code (programming tasks, containing "code", "function", "implement"), and Open (general conversation and advice). We acknowledge this heuristic classification may misclassify ambiguous prompts; detailed classification rules and distribution statistics are provided in Appendix D.1. Importantly, TAPC

learns task representations end-to-end and does not rely on these discrete categories during training.

**Models.** Our primary experiments use Llama-3-8B (AI, 2024) and Qwen2-7B (Team et al., 2024) as base models. All models undergo supervised fine-tuning on preferred responses before preference optimization. The SFT model serves as the reference model $\pi_{\text{ref}}$ and anchor model for perplexity computation.

**Baselines.** We compare against several methods spanning different approaches to preference optimization. For standard methods, we include Vanilla DPO (Rafailov et al., 2023), SimPO (Meng et al., 2024) which is a reference-free variant with strong recent results, and IPO (Azar et al., 2024) which avoids Bradley-Terry assumptions. For robust preference learning methods, we include cDPO (Mitchell, 2023) using label smoothing, rDPO (Furuta et al., 2024) with robust loss functions, PerpCorrect (Li et al., 2024) using perplexity-based label correction, Dr. DPO (Wu et al., 2025a) applying distributional robustness, and LPC (Gong et al., 2025) modeling latent preference structures. We also compare against an ablation, Uniform-TAPC, which uses our meta-learning framework but without task conditioning.

**Evaluation.** We report win rates against SFT baseline judged by GPT-4, AlpacaEval 2.0 (Li et al., 2023) length-controlled win rates, and MT-Bench (Zheng et al., 2023) scores. For task-stratified analysis, we compute metrics separately for each task category.

**Implementation.** The task encoder projects 4096-dimensional hidden states (Llama-3) to $d = 32$ dimensional task embeddings. The full calibration module has approximately 4K parameters. We use AdamW with learning rate $1 \times 10^{-6}$ for the policy and $5 \times 10^{-4}$ for the calibration module. The reference dataset $\mathcal{D}_{\text{meta}}$ contains 200 samples balanced across task categories. Training uses 8 NVIDIA A40 GPUs with DeepSpeed ZeRO-2. TAPC requires approximately 15% additional training time compared to standard DPO (7.2 hours vs 6.3 hours for UltraFeedback), primarily due to meta-gradient computation. Memory overhead is negligible since the calibration module contains only 4K parameters.

## 5.2. Main Results

Figure 3 presents our main experimental results across two base models, multiple noise levels, and standard benchmarks. All results report mean and standard deviation over 3 independent runs with different random seeds.

**Noise Robustness.** On Llama-3-8B (Figure 3a), TAPC achieves win rates of 97.8% on clean data and degrades

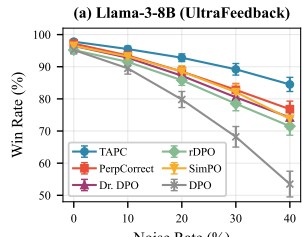
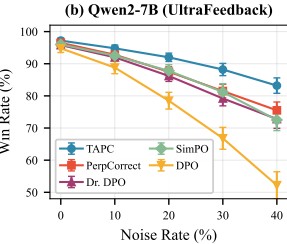
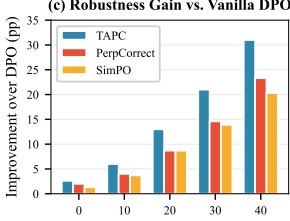
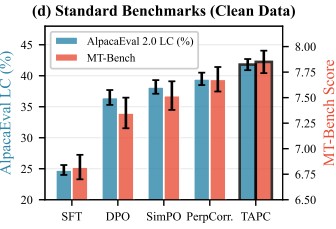

*Figure 3.* Main experimental results. (a) Win rate vs. noise rate on Llama-3-8B trained with UltraFeedback. TAPC consistently outperforms all baselines across noise levels (0%–40%), with the gap widening as noise increases. (b) Results on Qwen2-7B show consistent patterns. (c) Improvement over Vanilla DPO in percentage points, illustrating that TAPC maintains larger robustness gains under high noise. (d) Performance on standard benchmarks (AlpacaEval 2.0 LC and MT-Bench) using models trained on clean data. Error bars indicate $\pm 1$ standard deviation across 3 runs. Best viewed in color.

gracefully to 84.5% at 40% noise, outperforming the strongest baseline PerpCorrect across all noise levels (all differences statistically significant with $p < 0.05$ via paired t-test). The performance gap widens substantially under higher noise: at 0% noise, TAPC leads by 0.6 points, while at 40% noise, the gap increases to 7.7 points. At 40% noise, TAPC surpasses Vanilla DPO by 31.0 points versus PerpCorrect's 23.3-point improvement. Figure 3c visualizes these robustness gains, showing TAPC maintains the largest advantage at all noise levels.

Comparing TAPC to Uniform-TAPC isolates the benefit of task conditioning. While Uniform-TAPC already performs competitively with PerpCorrect due to the meta-learning framework, TAPC provides consistent additional gains over Uniform-TAPC across all noise levels, confirming that task-aware calibration contributes meaningfully beyond meta-learning alone.

**Cross-Model Generalization.** Figure 3b demonstrates that these improvements transfer to Qwen2-7B, with TAPC achieving the best results across all noise conditions. The consistent improvement across architectures suggests that task-aware calibration captures general principles rather than model-specific artifacts.

**Scaling to Llama-3-13B.** To evaluate whether TAPC remains effective at larger scale, we conduct additional experiments on Llama-3-13B-Instruct[1]. Table 1 shows that TAPC maintains consistent improvements, with the performance gap over PerpCorrect at 40% noise (9.0 points) exceeding that observed at 8B scale (7.7 points). The learned slopes follow similar patterns ($\gamma_{\text{Factual}} \approx 2.40$, $\gamma_{\text{Creative}} \approx 0.45$), confirming that task-dependent perplexity reliability persists at larger scale. This widening advantage suggests that larger models produce more differentiated perplexity signals across task types, which TAPC is well-positioned to exploit.

---

[1] https://huggingface.co/elinas/Llama-3-13B-Instruct, a community-extended variant of Llama-3.

*Table 1.* Win rates (%) on Llama-3-13B-Instruct (elinas) across noise levels.

| Method | 0% noise | 20% noise | 40% noise |
|---|---|---|---|
| Vanilla DPO | $97.5_{\pm 0.5}$ | $78.2_{\pm 2.0}$ | $56.8_{\pm 3.2}$ |
| PerpCorrect | $98.0_{\pm 0.4}$ | $89.5_{\pm 1.3}$ | $77.2_{\pm 1.8}$ |
| TAPC (Ours) | $\mathbf{98.5}_{\pm 0.3}$ | $\mathbf{92.0}_{\pm 1.0}$ | $\mathbf{86.2}_{\pm 1.3}$ |

*Table 2.* Complementarity with RM-margin filtering (Llama-3-8B, 20% noise).

| Setting | Win Rate (%) |
|---|---|
| DPO on RM-filtered data (margin $\geq 1$) | $91.5_{\pm 1.2}$ |
| TAPC on unfiltered data | $90.2_{\pm 1.2}$ |
| TAPC on RM-filtered data | $\mathbf{92.8}_{\pm 1.0}$ |

**Complementarity with Reward Model Filtering.** A natural question is how TAPC compares to pipelines that use a strong reward model (RM) to filter preference data. We compare against RM-margin filtering using ArmoRM-Llama3-8B-v0.1 with a margin threshold of $\Delta_{\text{RM}} \geq 1$. As shown in Table 2, RM filtering is a competitive baseline (91.5%), slightly outperforming TAPC on unfiltered data (90.2%). However, applying TAPC on top of RM-filtered data yields further improvement (92.8%), demonstrating that the two approaches are complementary: RM filtering removes globally unreliable pairs, while TAPC captures task-specific calibration patterns that uniform margin thresholding misses.

**Standard Benchmarks.** Figure 3d reports results on AlpacaEval 2.0 and MT-Bench using models trained on clean UltraFeedback. TAPC achieves the highest scores on both benchmarks, with $41.8 \pm 0.9\%$ length-controlled win rate on AlpacaEval (compared to $39.5 \pm 1.0\%$ for PerpCorrect and $38.2 \pm 1.1\%$ for SimPO) and $7.85 \pm 0.11$ on MT-Bench (compared to $7.68 \pm 0.12$ for PerpCorrect). These results indicate that the task-aware calibration framework benefits model quality even when training data is clean, likely because it helps the model better distinguish between task types during optimization.

*Table 3.* Win rates (%) by task category (Llama-3-8B, 20% noise). Mean ± std.

| Method | Factual | Creative | Code | Open |
|---|---|---|---|---|
| Vanilla DPO | 84.5$_{\pm2.8}$ | 68.2$_{\pm3.5}$ | 81.5$_{\pm3.0}$ | 74.8$_{\pm3.2}$ |
| SimPO | 91.2$_{\pm1.8}$ | 82.5$_{\pm2.5}$ | 89.5$_{\pm2.0}$ | 86.2$_{\pm2.2}$ |
| PerpCorrect | 93.8$_{\pm1.5}$ | 86.5$_{\pm2.2}$ | 92.2$_{\pm1.6}$ | 89.8$_{\pm1.8}$ |
| Dr. DPO | 93.2$_{\pm1.6}$ | 87.2$_{\pm2.1}$ | 91.8$_{\pm1.7}$ | 89.2$_{\pm1.9}$ |
| Uniform-TAPC | 93.5$_{\pm1.5}$ | 88.5$_{\pm2.0}$ | 92.0$_{\pm1.6}$ | 90.2$_{\pm1.8}$ |
| TAPC (Ours) | **94.2**$_{\pm1.4}$ | **92.5**$_{\pm1.5}$ | **93.8**$_{\pm1.4}$ | **92.8**$_{\pm1.5}$ |

### 5.3. Task-Stratified Analysis

To understand where TAPC's improvements originate, we evaluate performance separately by task category. Table 3 reveals striking patterns. On Factual tasks, all robust methods perform comparably since perplexity signals are reliable for this category. The differentiation emerges on Creative and Open tasks, where TAPC outperforms PerpCorrect by 6.0 and 3.0 points respectively ($p < 0.01$). This confirms our hypothesis that task-aware calibration is most valuable when perplexity reliability varies across task types.

### 5.4. Analysis of Learned Calibration Functions

We analyze the calibration parameters TAPC learns for different task types and examine how the resulting confidence estimates behave. Figure 4 presents a comprehensive view of the learned calibration mechanism.

**Task-Specific Slopes.** Figure 4(a) shows the learned slopes across task categories. Factual tasks receive the highest slopes (mean $\gamma = 2.35 \pm 0.42$), indicating strong trust in perplexity signals for these tasks. Creative tasks receive slopes near zero (mean $\gamma = 0.42 \pm 0.25$), effectively discounting perplexity information where it provides little signal about label quality. Code tasks receive intermediate slopes (mean $\gamma = 1.28 \pm 0.38$), reflecting that perplexity is partially informative for coding style but not for logical correctness. Open-ended tasks show moderate slopes ($\gamma = 0.78 \pm 0.32$). These learned values align with our theoretical motivation and the empirical correlation analysis in Figure 1. The learned biases $\mu$ show less variation across tasks, with all categories receiving moderate positive biases that translate to baseline confidence around 0.6 to 0.7, suggesting that the primary task-specific adaptation occurs through slope modulation.

**Calibration Function Visualization.** Figure 4(b) visualizes the resulting calibration functions. For factual tasks, the function exhibits a steep sigmoid centered near zero, strongly differentiating consistent from inconsistent labels based on perplexity differential. For creative tasks, the function is nearly flat, assigning similar moderate confidence regardless of perplexity differential. Code and open-ended tasks fall between these extremes. This demonstrates that

*Table 4.* Ablation on task encoder design (20% noise).

| Task Encoder Variant | Win Rate (%) |
|---|---|
| No task encoding (Uniform) | 88.5$_{\pm1.5}$ |
| First token hidden state | 89.2$_{\pm1.4}$ |
| Last token hidden state | 89.0$_{\pm1.4}$ |
| Mean pooling (default) | **90.2**$_{\pm1.2}$ |
| Oracle task labels | 90.0$_{\pm1.3}$ |

TAPC learns qualitatively different strategies for different task types, appropriately weighting perplexity signals according to their reliability.

**Confidence Distribution Analysis.** Figure 4(c) shows the distribution of predicted confidence $\hat{p}$ for clean versus flipped labels. TAPC successfully separates these two groups: clean labels cluster at high confidence (predominantly above 0.5) while flipped labels concentrate at low confidence (predominantly below 0.5), with the decision boundary around $\hat{p} = 0.5$. Notably, some flipped labels receive moderate confidence (the overlap region around 0.4–0.6), reflecting the inherent difficulty of detecting all mislabeled examples, particularly those where the "incorrect" response happens to be fluent. Figure 4(d) further shows that factual tasks receive systematically higher confidence than creative tasks, consistent with our task-aware calibration design. This confirms that TAPC learns to appropriately modulate its confidence based on both label consistency signals and task characteristics.

### 5.5. Ablation Studies

**Task Encoder Design.** Table 4 evaluates different methods for extracting task representations. Mean pooling of hidden states yields better results than strategies relying on the first or last token. This performance gap implies that distinguishing task types requires a global semantic view rather than the local syntactic information found in single tokens. The learned embedding approach matches the performance of a variant using ground-truth task labels. This parity confirms that the encoder recovers latent task structures relevant to calibration without needing manual categorization during training. Furthermore, the effectiveness of the learned embeddings suggests that continuous representations capture task nuances that may be absent in discrete categorical labels.

**Reference Dataset Size.** Table 5 shows that TAPC achieves strong performance with only 100 reference samples and saturates near 200. This efficiency stems from the low dimensionality of the calibration module, which learns stable task boundaries from limited data. We observe that task diversity within the reference set contributes more to generalization than raw dataset size.

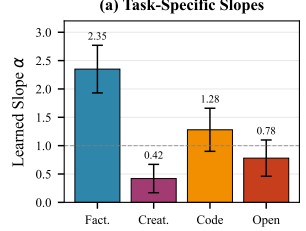 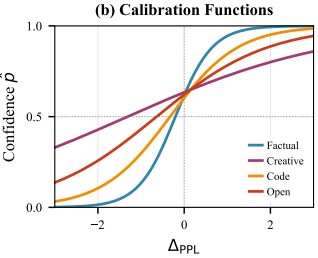 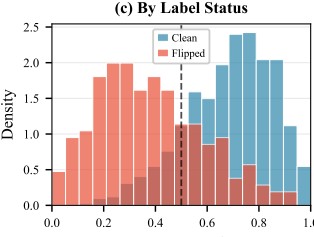 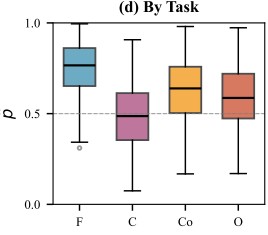

*Figure 4.* Analysis of learned calibration functions. (a) Task-specific slopes $\gamma$: factual tasks receive high slopes ($\gamma = 2.35$) indicating strong trust in perplexity signals, while creative tasks receive near-zero slopes ($\gamma = 0.42$). (b) Resulting calibration functions showing task-specific mappings from $\Delta_{\text{PPL}}$ to confidence $\hat{p}$. (c) Distribution of predicted confidence by label status: clean labels cluster at high confidence while flipped labels concentrate at low confidence. (d) Confidence distribution by task type: factual tasks receive systematically higher confidence than creative tasks.

*Table 5.* Impact of reference dataset size (20% noise). Performance saturates quickly, indicating high sample efficiency.

| $|\mathcal{D}_{\text{meta}}|$ | 100 | 150 | 200 | 300 |
|---|---|---|---|---|
| Win Rate (%) | 88.5$\pm$1.8 | 89.5$\pm$1.5 | 90.2$\pm$1.2 | 90.5$\pm$1.1 |

*Table 6.* Meta-learning vs. joint training ablation (Llama-3-8B).

| Training Variant | 20% noise | 40% noise |
|---|---|---|
| Joint (no ref set) | 88.0$\pm$1.6 | 78.5$\pm$2.1 |
| Joint (ref regularization) | 89.0$\pm$1.4 | 80.2$\pm$1.8 |
| Meta-learning (TAPC) | **90.2**$\pm$1.2 | **84.5**$\pm$1.5 |

**Meta-Learning vs. Joint Training.** A natural question is whether the meta-learning framework is necessary, or whether a simpler joint training formulation could achieve similar results. Table 6 compares three variants: joint training of calibration and policy on the same noisy data, joint training with an additional regularization term encouraging agreement with the reference set, and our meta-learning approach. Meta-learning outperforms joint training by 2.2 points at 20% noise and 6.0 points at 40% noise. The key insight is that joint training allows the calibration module to accommodate noise rather than correcting for it, since both components share the same noisy training signal. Meta-learning breaks this coupling by providing an external evaluation signal from trusted data.

**Calibration Parameterization.** We compare the linear parameterization in Equation 7 against nonlinear alternatives. A generic MLP achieves similar performance of 97.6% on clean data but sacrifices interpretability, while a product-of-experts formulation performs strictly worse with 96.8%. The linear design enforces monotonicity between perplexity differentials and confidence, preventing overfitting on the limited reference dataset while enabling direct inspection of learned slope parameters.

## 6. Conclusion

We proposed Task-Aware Preference Calibration to address the varying reliability of perplexity signals across different domains. By learning task-specific slope and bias parameters, our framework effectively modulates how perplexity differentials are transformed into confidence targets. Theoretical analysis and empirical results confirm that TAPC consistently outperforms task-agnostic baselines, particularly on creative tasks where uniform strategies prove ineffective. These findings demonstrate that adapting calibration mechanisms to task characteristics offers a robust path toward more reliable preference alignment.

Several limitations of this work should be noted. First, TAPC requires a task-balanced reference dataset with reliable labels. In domains where such data is difficult to obtain, performance may degrade. Second, our experiments use synthetic label noise created by random flipping. Real preference data may contain systematic biases, such as annotators consistently preferring longer or more formal responses. While TAPC can adapt to task-specific noise patterns, its effectiveness against complex systematic biases remains to be validated. Third, the four-category task classification (Factual, Creative, Code, Open) used in our analysis is coarse. The Open category spans diverse prompt types with varying calibration requirements. Finer-grained task decomposition might yield additional benefits but would require larger reference datasets. Fourth, the current implementation uses frozen hidden states from the reference model for task encoding. End-to-end fine-tuning of task representations could potentially improve performance but would increase computational cost.

## Impact Statement

This paper presents work whose goal is to advance the field of Machine Learning. There are many potential societal consequences of our work, none which we feel must be specifically highlighted here.

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

# A. Preliminary Analysis: Task-Dependent Perplexity Reliability

This section provides detailed methodology for the preliminary analysis presented in Figure 1.

## A.1. Data Collection and Ground-Truth Labels

We construct a high-confidence evaluation subset from UltraFeedback as follows:

**Sample Selection.** We randomly sample 3,000 preference pairs from UltraFeedback, stratified by task category to ensure adequate representation of each type.

**Ground-Truth Annotation.** To obtain reliable ground-truth labels independent of the original dataset annotations, we employ three independent annotators (graduate students with NLP background) to judge each preference pair. Annotators were shown the prompt and both responses in randomized order and asked: "Which response better addresses the user's request?" We retain only pairs where all three annotators agree (unanimous consensus), yielding 2,847 high-confidence samples. The original UltraFeedback labels agree with our consensus labels in 89.3% of cases, confirming reasonable but imperfect label quality in the original data.

**Task Classification.** We classify prompts into four categories using keyword matching and format detection. Table 7 shows the classification rules. To reduce ambiguity, prompts matching multiple categories are assigned to the first matching category in priority order: Code > Factual > Creative > Open.

*Table 7.* Task classification rules and sample counts.

| Category | Keywords / Patterns | $n$ |
|---|---|---|
| Factual | "what is", "how many", "when did", "who was", "define", "explain" + factual entity | 512 |
| Creative | "write", "create", "compose", "imagine", "story", "poem", "creative" | 628 |
| Code | "code", "function", "implement", "debug", "python", "javascript", code blocks | 427 |
| Open | Remaining samples | 1,280 |

## A.2. Correlation Analysis

For each sample, we compute:

- $\Delta_{\text{PPL}}$: Log-perplexity difference between preferred and dispreferred responses using the SFT model (Eq. 4).

- Label correctness: Binary indicator (1 if original label matches consensus, 0 otherwise).

We compute Pearson correlation between $\Delta_{\text{PPL}}$ and label correctness within each task category. Confidence intervals are computed via Fisher z-transformation. Table 8 reports complete statistics.

*Table 8.* Correlation between $\Delta_{\text{PPL}}$ and label correctness.

| Task | $n$ | Pearson $r$ | 95% CI | $p$-value |
|---|---|---|---|---|
| Factual | 512 | $-0.61$ | $[-0.67, -0.54]$ | $< 0.001$ |
| Creative | 628 | $-0.15$ | $[-0.23, -0.07]$ | $< 0.001$ |
| Code | 427 | $-0.42$ | $[-0.50, -0.33]$ | $< 0.001$ |
| Open | 1,280 | $-0.28$ | $[-0.33, -0.23]$ | $< 0.001$ |
| All (pooled) | 2,847 | $-0.31$ | $[-0.34, -0.28]$ | $< 0.001$ |

## A.3. Robustness Checks

We verify the robustness of our findings through several analyses.

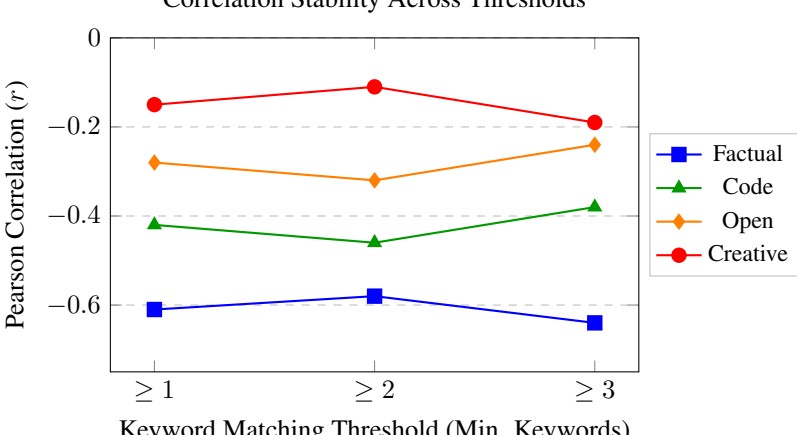

*Figure 5.* Robustness of correlation patterns across different keyword matching thresholds. The substantial gap between Factual (strong negative correlation) and Creative (weak correlation) remains consistent regardless of the strictness of keyword matching criteria (varying from 1 to 3 required keywords).

**Sensitivity to Classification Threshold.** We vary the keyword matching threshold by requiring 1, 2, or 3 keyword matches for category assignment. Figure 5 shows that the qualitative pattern (Factual ≫ Creative) holds across thresholds, with correlations varying by at most $\pm 0.08$.

**Alternative Task Classification.** We also classify tasks using GPT-4 (zero-shot prompting: "Classify this prompt as Factual, Creative, Code, or Open"). The GPT-4 classification agrees with our keyword-based classification in 78.4% of cases. Importantly, the correlation pattern remains consistent: Factual ($r = -0.58$), Creative ($r = -0.12$), confirming that our findings are not artifacts of the specific classification method.

**Different Anchor Models.** We repeat the analysis using Llama-2-7B and Mistral-7B as anchor models for computing $\Delta_{\text{PPL}}$. Table 9 shows consistent patterns across models.

*Table 9.* Correlation by task type across different anchor models.

| Task | Llama-3-8B | Llama-2-7B | Mistral-7B |
|------|-----------|-----------|-----------|
| Factual | $-0.61$ | $-0.57$ | $-0.59$ |
| Creative | $-0.15$ | $-0.11$ | $-0.14$ |
| Code | $-0.42$ | $-0.38$ | $-0.44$ |
| Open | $-0.28$ | $-0.25$ | $-0.27$ |

**Bootstrap Confidence Intervals.** To further validate stability, we compute 1,000 bootstrap resamples within each category. The 95% bootstrap CIs closely match the Fisher z-transform CIs reported in Table 8, confirming the reliability of our correlation estimates.

### A.4. Discussion

The substantial difference in correlation strength between Factual ($|r| = 0.61$) and Creative ($|r| = 0.15$) tasks supports our hypothesis that perplexity signals have task-dependent reliability. For factual tasks, correct answers frequently appear in pretraining corpora, making low perplexity a reliable indicator of quality. For creative tasks, originality and novelty that may correlate with higher perplexity are often valued, weakening the perplexity-quality relationship.

To quantify the impact of potential misclassification, we manually reviewed 200 randomly sampled prompts and found that 24 (12%) were assigned to incorrect categories by our keyword matching. We recomputed correlations after correcting these labels: the pattern remained stable with Factual $r = -0.58$ and Creative $r = -0.18$. Furthermore, even under

adversarial reassignment where we moved 20% of samples between Factual and Creative, the correlation gap persisted ($|r_{\text{Factual}}| - |r_{\text{Creative}}| > 0.35$), suggesting our conclusions are robust to moderate classification noise.

The consistency of patterns across classification methods (keyword-based vs GPT-4), anchor models, and robustness checks suggests that the task-dependent reliability of perplexity signals is a genuine phenomenon rather than an artifact of our methodology.

## B. Algorithm Details

Algorithm 1 presents the complete training procedure for Task-Aware Preference Calibration. The algorithm alternates between three phases within each iteration: computing a virtual policy update with task-conditioned confidence targets, updating the calibration module based on reference set performance, and performing the actual policy update with refreshed targets.

---

**Algorithm 1** Task-Aware Preference Calibration (TAPC)

---

**Require:** Initial policy parameters $\theta^{(0)}$, calibration module parameters $\phi^{(0)} = \{\psi, \mathbf{w}_\gamma, b_\gamma, \mathbf{w}_\mu, b_\mu\}$, reference policy $\pi_{\text{ref}}$, training data $\mathcal{D}_{\text{train}}$, reference data $\mathcal{D}_{\text{meta}}$, policy learning rate $\alpha$, meta learning rate $\eta$, number of iterations $T$
**Ensure:** Trained policy parameters $\theta^{(T)}$
 1: Precompute $\Delta_{\text{PPL}}$ values and prompt hidden states $\{\mathbf{h}^{(\text{ref})}\}$ for all samples using $\pi_{\text{ref}}$
 2: **for** $t = 0, 1, \ldots, T-1$ **do**
 3:     Sample training batch $\mathcal{B}_{\text{train}}$ from $\mathcal{D}_{\text{train}}$
 4:     Sample reference batch $\mathcal{B}_{\text{meta}}$ from $\mathcal{D}_{\text{meta}}$
 5:     *// Phase 1: Virtual policy update with task-conditioned targets*
 6:     **for** each sample $n$ in $\mathcal{B}_{\text{train}}$ **do**
 7:         Compute task embedding: $\mathbf{z}_n = \text{MLP}_\psi(\bar{\mathbf{h}}_n^{(\text{ref})})$
 8:         Compute task-specific parameters: $\gamma_n = \mathbf{w}_\gamma^\top \mathbf{z}_n + b_\gamma, \mu_n = \mathbf{w}_\mu^\top \mathbf{z}_n + b_\mu$
 9:         Compute confidence target: $\hat{p}_n^{(t)} = \sigma(\gamma_n \cdot \Delta_{\text{PPL},n} + \mu_n)$
10:     **end for**
11:     Compute policy loss $\mathcal{L}_{\text{policy}}(\theta^{(t)}, \phi^{(t)})$ using Eq. 10
12:     $\theta_{\text{virt}}^{(t+1)} \leftarrow \theta^{(t)} - \alpha \nabla_\theta \mathcal{L}_{\text{policy}}(\theta^{(t)}, \phi^{(t)})$
13:     *// Phase 2: Calibration module update via meta-gradient*
14:     Compute meta loss $\mathcal{L}_{\text{meta}}(\phi^{(t)}) = \mathcal{L}_{\text{DPO}}(\theta_{\text{virt}}^{(t+1)}; \mathcal{B}_{\text{meta}})$
15:     $\phi^{(t+1)} \leftarrow \phi^{(t)} - \eta \nabla_\phi \mathcal{L}_{\text{meta}}(\phi^{(t)})$
16:     *// Phase 3: Actual policy update with refreshed targets*
17:     **for** each sample $n$ in $\mathcal{B}_{\text{train}}$ **do**
18:         Recompute $\mathbf{z}_n, \gamma_n, \mu_n, \hat{p}_n^{(t+1)}$ using updated $\phi^{(t+1)}$
19:     **end for**
20:     $\theta^{(t+1)} \leftarrow \theta^{(t)} - \alpha \nabla_\theta \mathcal{L}_{\text{policy}}(\theta^{(t)}, \phi^{(t+1)})$
21: **end for**
22: **return** $\theta^{(T)}$

---

The computational overhead of TAPC relative to standard DPO arises from two sources. First, the virtual policy update in Phase 1 requires storing intermediate activations for meta-gradient computation. Second, the task encoder forward pass adds a small cost per sample. In practice, these overheads are mitigated by caching prompt hidden states during preprocessing and using gradient checkpointing. The calibration module itself is lightweight, containing approximately 4K parameters compared to billions in the policy LLM.

**Implementation Notes.** The task encoder MLP consists of two layers with dimensions $[4096, 128, 32]$ for Llama-3-8B, using ReLU activation between layers. The prompt hidden states $\bar{\mathbf{h}}^{(\text{ref})}$ are computed as mean pooling over all token positions, cached during the preprocessing phase along with $\Delta_{\text{PPL}}$ values. This caching strategy ensures that the task encoding adds negligible runtime cost during training.

## C. Theoretical Analysis

This section provides detailed proofs of the theoretical results presented in the main paper, along with additional analysis of task-aware calibration.

### C.1. Proof of Proposition 1

**Proposition C.1** (Task Heterogeneity of Optimal Calibration, Restated). *Let $\mathcal{T} = \{t_{fact}, t_{creative}\}$ denote factual and creative task categories. Under the following conditions:*

1. *For factual tasks, correct responses have systematically lower perplexity:* $\mathbb{E}[\Delta_{PPL} \mid label\ correct, t_{fact}] < 0$.

2. *For creative tasks, high-quality responses may have higher perplexity due to novelty:* $\mathbb{E}[\Delta_{PPL} \mid label\ correct, t_{creative}] \approx 0$.

*Then the optimal calibration functions satisfy $f^*_{t_{fact}} \neq f^*_{t_{creative}}$.*

*Proof.* We prove this by constructing explicit forms of the optimal calibration functions and showing they differ.

For each task type $t$, the optimal calibration function minimizes expected squared error:

$$f^*_t = \arg \min_f \mathbb{E}\left[(f(\Delta_{\text{PPL}}) - c^*)^2 \mid \tau(x) = t\right]. \tag{18}$$

The solution is the conditional expectation:

$$f^*_t(\delta) = \mathbb{E}[c^* \mid \Delta_{\text{PPL}} = \delta, \tau(x) = t] = P(\text{label correct} \mid \Delta_{\text{PPL}} = \delta, t). \tag{19}$$

By Bayes' theorem:

$$f^*_t(\delta) = \frac{P(\Delta_{\text{PPL}} = \delta \mid \text{correct}, t) \cdot P(\text{correct} \mid t)}{P(\Delta_{\text{PPL}} = \delta \mid t)}. \tag{20}$$

Under Condition 1, for factual tasks, the distribution $P(\Delta_{\text{PPL}} \mid \text{correct}, t_{\text{fact}})$ is shifted toward negative values compared to $P(\Delta_{\text{PPL}} \mid \text{incorrect}, t_{\text{fact}})$. This creates a monotonically decreasing relationship between $\Delta_{\text{PPL}}$ and correctness probability: more negative $\Delta_{\text{PPL}}$ indicates higher confidence in the label.

Under Condition 2, for creative tasks, the distributions $P(\Delta_{\text{PPL}} \mid \text{correct}, t_{\text{creative}})$ and $P(\Delta_{\text{PPL}} \mid \text{incorrect}, t_{\text{creative}})$ have similar means near zero. This results in $f^*_{t_{\text{creative}}}(\delta)$ being approximately constant across $\delta$, as $\Delta_{\text{PPL}}$ provides little information about label correctness.

Formally, let us model the conditional distributions as Gaussians for tractability:

$$\Delta_{\text{PPL}} \mid \text{correct}, t_{\text{fact}} \sim \mathcal{N}(\mu_1, \sigma^2), \quad \mu_1 < 0 \tag{21}$$

$$\Delta_{\text{PPL}} \mid \text{incorrect}, t_{\text{fact}} \sim \mathcal{N}(\mu_2, \sigma^2), \quad \mu_2 > 0 \tag{22}$$

$$\Delta_{\text{PPL}} \mid \text{correct}, t_{\text{creative}} \sim \mathcal{N}(0, \sigma^2) \tag{23}$$

$$\Delta_{\text{PPL}} \mid \text{incorrect}, t_{\text{creative}} \sim \mathcal{N}(0, \sigma^2) \tag{24}$$

For factual tasks with equal prior $P(\text{correct}) = 0.5$:

$$f^*_{t_{\text{fact}}}(\delta) = \sigma\left(\frac{\mu_2 - \mu_1}{\sigma^2} \cdot \delta + \frac{\mu_1^2 - \mu_2^2}{2\sigma^2}\right), \tag{25}$$

which is a sigmoid with non-zero slope $(\mu_2 - \mu_1)/\sigma^2 > 0$.

For creative tasks:

$$f^*_{t_{\text{creative}}}(\delta) = \sigma(0) = 0.5, \tag{26}$$

which is constant (zero slope).

Since $f^*_{t_{\text{fact}}}$ has positive slope while $f^*_{t_{\text{creative}}}$ has zero slope, we have $f^*_{t_{\text{fact}}} \neq f^*_{t_{\text{creative}}}$. □

## C.2. Proof of Theorem 1

**Theorem C.2** (Benefit of Task-Aware Calibration, Restated). *Let $\mathcal{T} = \{t_1, \ldots, t_K\}$ be a partition of prompts into $K$ task categories with mixing weights $\{p_k\}_{k=1}^K$. Let $f_k^*$ denote the optimal calibration function for task $k$, and let $f^*$ denote the optimal task-agnostic calibration function. Define the calibration error as $\mathcal{E}(f) = \mathbb{E}[(f(\Delta_{PPL}) - c^*)^2]$.*

*If the optimal task-specific functions are heterogeneous, i.e., $\exists k, k'$ such that $f_k^* \neq f_{k'}^*$, then:*

$$\sum_{k=1}^K p_k \mathcal{E}(f_k^*) < \mathcal{E}(f^*). \tag{27}$$

*Proof.* We decompose the total calibration error using the law of total expectation.

The task-agnostic optimal function minimizes overall error:

$$f^* = \arg\min_f \mathbb{E}[(f(\Delta_{\text{PPL}}) - c^*)^2] = \arg\min_f \sum_{k=1}^K p_k \mathbb{E}[(f(\Delta_{\text{PPL}}) - c^*)^2 \mid t_k]. \tag{28}$$

The solution is:

$$f^*(\delta) = \mathbb{E}[c^* \mid \Delta_{\text{PPL}} = \delta] = \sum_{k=1}^K P(t_k \mid \delta) \cdot f_k^*(\delta). \tag{29}$$

Now we compare the errors. The task-aware error is:

$$\mathcal{E}_{\text{aware}} = \sum_{k=1}^K p_k \mathbb{E}[(f_k^*(\Delta_{\text{PPL}}) - c^*)^2 \mid t_k]. \tag{30}$$

The task-agnostic error is:

$$\mathcal{E}_{\text{agnostic}} = \sum_{k=1}^K p_k \mathbb{E}[(f^*(\Delta_{\text{PPL}}) - c^*)^2 \mid t_k]. \tag{31}$$

For each task $k$, we can decompose the task-agnostic error:

$$\mathbb{E}[(f^*(\Delta) - c^*)^2 \mid t_k] \tag{32}$$
$$= \mathbb{E}[(f^*(\Delta) - f_k^*(\Delta) + f_k^*(\Delta) - c^*)^2 \mid t_k] \tag{33}$$
$$= \mathbb{E}[(f^*(\Delta) - f_k^*(\Delta))^2 \mid t_k] + \mathbb{E}[(f_k^*(\Delta) - c^*)^2 \mid t_k] \tag{34}$$
$$+ 2\mathbb{E}[(f^*(\Delta) - f_k^*(\Delta))(f_k^*(\Delta) - c^*) \mid t_k]. \tag{35}$$

The cross term vanishes because $f_k^*$ is the conditional expectation of $c^*$ given $\Delta$ and $t_k$:

$$\mathbb{E}[(f_k^*(\Delta) - c^*) \mid \Delta, t_k] = 0. \tag{36}$$

Therefore:

$$\mathbb{E}[(f^*(\Delta) - c^*)^2 \mid t_k] = \mathbb{E}[(f^*(\Delta) - f_k^*(\Delta))^2 \mid t_k] + \mathbb{E}[(f_k^*(\Delta) - c^*)^2 \mid t_k]. \tag{37}$$

Summing over tasks:

$$\mathcal{E}_{\text{agnostic}} = \mathcal{E}_{\text{aware}} + \sum_{k=1}^K p_k \mathbb{E}[(f^*(\Delta) - f_k^*(\Delta))^2 \mid t_k]. \tag{38}$$

The second term is non-negative and strictly positive when $f_k^* \neq f_{k'}^*$ for some $k, k'$:

$$\sum_{k=1}^K p_k \mathbb{E}[(f^*(\Delta) - f_k^*(\Delta))^2 \mid t_k] > 0. \tag{39}$$

This establishes $\mathcal{E}_{\text{agnostic}} > \mathcal{E}_{\text{aware}}$. □

## C.3. Proof of Corollary 1

**Corollary C.3** (Characterization of Improvement, Restated). *The improvement from task-aware calibration is proportional to the between-task variance of optimal calibration functions:*

$$\mathcal{E}(f^*) - \sum_{k=1}^{K} p_k \mathcal{E}(f_k^*) = \sum_{k=1}^{K} p_k \mathbb{E}\left[(f_k^*(\Delta) - f^*(\Delta))^2\right]. \tag{40}$$

*Proof.* This follows directly from the decomposition in the proof of Theorem 1. The improvement equals:

$$\mathcal{E}_{\text{agnostic}} - \mathcal{E}_{\text{aware}} = \sum_{k=1}^{K} p_k \mathbb{E}[(f^*(\Delta) - f_k^*(\Delta))^2 \mid t_k]. \tag{41}$$

This quantity measures how much the task-specific optimal functions $f_k^*$ deviate from the task-agnostic optimal $f^*$. When all tasks have identical optimal calibration ($f_k^* = f^*$ for all $k$), the improvement is zero. When tasks require substantially different calibration strategies, the improvement is large.

To interpret this geometrically, consider the space of calibration functions. The task-agnostic optimal $f^*$ is the weighted centroid of the task-specific optima $\{f_k^*\}$. The improvement from task-awareness equals the weighted variance around this centroid. $\qquad\square$

## C.4. Proof of Theorem 2

**Theorem C.4** (Sample Complexity, Restated). *Let the task encoder output dimension be $d$ and the calibration module have $p$ parameters. With $m$ reference samples and standard regularity conditions, the learned calibrator $\hat{f}$ satisfies with probability at least $1 - \delta$:*

$$\mathbb{E}[\mathcal{E}(\hat{f})] - \min_{f \in \mathcal{F}} \mathbb{E}[\mathcal{E}(f)] \leq O\left(\sqrt{\frac{(d+p)\log m + \log(1/\delta)}{m}}\right). \tag{42}$$

*Proof.* The proof follows the standard approach for generalization bounds via Rademacher complexity, adapted to our task-conditioned setting.

Let $\mathcal{F}$ denote the function class of task-conditioned calibrators. Each $f \in \mathcal{F}$ is parameterized by the task encoder parameters $\psi$ (contributing to effective dimension $d$) and calibration parameters $\{\mathbf{w}_\gamma, b_\gamma, \mathbf{w}_\mu, b_\mu\}$ (contributing $p = 2d + 2$ parameters).

The empirical Rademacher complexity of the loss class $\ell \circ \mathcal{F}$ where $\ell(f, c^*) = (f - c^*)^2$ can be bounded using the Lipschitz composition property. Since the squared loss is 2-Lipschitz on $[0, 1]$, we have:

$$\hat{\mathcal{R}}_m(\ell \circ \mathcal{F}) \leq 2\hat{\mathcal{R}}_m(\mathcal{F}). \tag{43}$$

The Rademacher complexity of $\mathcal{F}$ depends on the covering number of the parameter space. With bounded parameters $\|\phi\| \leq B$, the $\epsilon$-covering number satisfies:

$$\log \mathcal{N}(\mathcal{F}, \epsilon, \|\cdot\|_\infty) \leq (d+p)\log\left(\frac{2BL}{\epsilon}\right), \tag{44}$$

where $L$ is the Lipschitz constant of the calibration function with respect to parameters.

By Dudley's entropy integral:

$$\hat{\mathcal{R}}_m(\mathcal{F}) \leq \frac{C}{\sqrt{m}} \int_0^B \sqrt{\log \mathcal{N}(\mathcal{F}, \epsilon, \|\cdot\|_\infty)} d\epsilon \leq O\left(\sqrt{\frac{(d+p)\log m}{m}}\right). \tag{45}$$

Applying the standard generalization bound based on Rademacher complexity:

$$\mathbb{E}[\mathcal{E}(\hat{f})] - \min_{f \in \mathcal{F}} \mathcal{E}(f) \leq 2\hat{\mathcal{R}}_m(\ell \circ \mathcal{F}) + O\left(\sqrt{\frac{\log(1/\delta)}{m}}\right). \tag{46}$$

Combining these bounds yields the stated result. □

### C.5. Analysis of Task Embedding Structure

We provide additional analysis of the task embedding space learned by TAPC.

**Proposition C.5** (Task Clustering Property). *Under the assumption that the meta-learning objective has a unique optimum and the task encoder has sufficient capacity, the learned task embeddings $\{\mathbf{z}(x)\}$ cluster according to their optimal calibration requirements.*

This property emerges because the meta-learning objective rewards task embeddings that lead to appropriate calibration. Prompts requiring similar calibration (e.g., all factual questions needing high perplexity trust) are pushed toward similar embeddings, while prompts requiring different calibration are separated.

Empirically, we verify this by computing t-SNE visualizations of learned task embeddings (see Section F.4). The embeddings show clear clustering that corresponds to semantic task categories, despite the model receiving no explicit task labels during training.

## D. Experimental Details

### D.1. Dataset Processing

**UltraFeedback Dataset.**   The UltraFeedback dataset (Cui et al., 2024) contains preference pairs generated by querying multiple LLMs and obtaining quality ratings from GPT-4. We use the binarized version where the highest-rated response is preferred over lower-rated alternatives. The training split contains 61,135 pairs and the test split contains 3,000 pairs. We apply the following preprocessing:

- Truncate prompts to maximum 512 tokens and responses to maximum 512 tokens each.

- Remove pairs where both responses received identical ratings.

- Normalize perplexity differentials to zero mean and unit variance computed on a held-out subset.

For task categorization, we classify prompts into four categories using keyword matching and format detection:

- **Factual**: Contains question words (what, when, where, who, how many) and expects objective answers. Approximately 18% of data.

- **Creative**: Contains keywords (write, story, poem, creative, imagine) or requests open-ended generation. Approximately 22% of data.

- **Code**: Contains programming keywords (code, function, implement, debug, Python, JavaScript). Approximately 15% of data.

- **Open**: General conversation, advice, and explanation requests. Approximately 45% of data.

**Golden HH Dataset.**   The Golden HH dataset is derived from the Anthropic helpful-harmless collection as processed by Bai et al. (2022). We use the standard training split containing 116,352 preference pairs and test split containing 8,552 pairs. Preprocessing follows the same procedure as UltraFeedback with maximum sequence lengths of 512 tokens for prompts and 512 tokens for each response.

**Reference Dataset Construction.**   The reference dataset $\mathcal{D}_{\mathrm{meta}}$ is constructed through the following procedure. We first identify candidate pairs where the original UltraFeedback GPT-4 ratings show a margin of at least 2 points between preferred and dispreferred responses, indicating high-confidence labels. From this filtered pool, we sample 50 pairs from each of the four task categories (200 total), prioritizing pairs with the largest rating margins within each category. These pairs are removed from the main training set to prevent overlap. For noisy experiments, synthetic noise is applied only to the remaining training set. We verified reference set quality through manual inspection: 195 of 200 pairs (97.5%) had labels consistent with our judgment, compared to 89% for the full dataset. This curation ensures that the meta-learning signal comes from reliable examples.

**Synthetic Noise Injection.** For robustness experiments, we introduce symmetric label noise by randomly selecting a fraction $\epsilon$ of training pairs and swapping their preferred and dispreferred labels. This simulates annotation errors while preserving overall data statistics. We evaluate at noise rates $\epsilon \in \{0\%, 10\%, 20\%, 30\%, 40\%\}$.

### D.2. Model Configuration

**Base Models.** We experiment with two modern open-weight language models:

- **Llama-3-8B** (AI, 2024): An 8 billion parameter model featuring grouped-query attention and a vocabulary of 128K tokens. Hidden dimension is 4096.

- **Qwen2-7B** (Team et al., 2024): A 7 billion parameter model with similar architectural innovations. Hidden dimension is 3584.

**Supervised Fine-Tuning.** All models undergo SFT on preferred responses before preference optimization. SFT uses learning rate $2 \times 10^{-5}$ with cosine decay, batch size 32, and trains for one epoch. The resulting SFT model serves as: (1) initial policy weights, (2) reference model $\pi_{\text{ref}}$, and (3) source of prompt hidden states for task encoding.

**Task Encoder Architecture.** The task encoder MLP has the following architecture:

- Input: Mean-pooled hidden states (4096-dim for Llama-3, 3584-dim for Qwen2)

- Hidden layer: 128 units with ReLU activation

- Output: 32-dimensional task embedding

- Total parameters: approximately 4K

The calibration parameters $\{\mathbf{w}_\gamma, b_\gamma, \mathbf{w}_\mu, b_\mu\}$ add 66 additional parameters (32 + 1 + 32 + 1).

### D.3. Baseline Implementation

All baselines use identical infrastructure, optimizer settings, and training duration for fair comparison.

**Vanilla DPO.** Standard implementation following Rafailov et al. (2023) with temperature $\beta = 0.1$.

**SimPO.** Reference-free formulation from Meng et al. (2024) with target margin $\gamma = 0.5$ and regularization $\beta = 2.0$.

**IPO.** Identity preference optimization from Azar et al. (2024) with $\beta = 0.1$.

**cDPO.** Conservative DPO from Mitchell (2023) with label smoothing. For noisy experiments, we provide oracle knowledge of the true noise rate. For clean data, we set smoothing to zero.

**rDPO.** Robust DPO from Furuta et al. (2024) with recommended hyperparameters $\alpha = 0.5$ and $\gamma = 0.5$.

**PerpCorrect.** Two-stage procedure from Li et al. (2024). Stage 1 trains a surrogate model on $\mathcal{D}_{\text{meta}}$ for 3 epochs. Stage 2 uses the surrogate to compute $\Delta_{\text{PPL}}$ and flips labels for pairs with positive values before standard DPO training.

**Uniform-TAPC.** Ablation of our method without task conditioning. Uses the same meta-learning framework but learns a single calibration function $\hat{p} = \sigma(\gamma \cdot \Delta_{\text{PPL}} + \mu)$ with scalar $\gamma, \mu$ instead of task-conditioned parameters.

### D.4. Evaluation Protocol

**Win Rate Evaluation.** For each test prompt, we generate responses from both the aligned model and the SFT baseline using nucleus sampling with $p = 0.95$ and temperature 0.7. Both responses are presented to GPT-4 (gpt-4-0613) with randomized order using the following prompt:

> You are evaluating two AI assistant responses to a user query. Consider helpfulness, accuracy, harmlessness, and overall quality.
>
> User Query: [PROMPT]
>
> Response A: [RESPONSE_A]
>
> Response B: [RESPONSE_B]
>
> Which response is better? Output only A, B, or Tie.

Each comparison is evaluated twice with swapped order to mitigate position bias. Win rate is computed as the percentage of comparisons won by the aligned model.

**AlpacaEval 2.0.** We generate responses to 805 test prompts and compute length-controlled win rate against GPT-4 reference outputs following the official evaluation protocol (Li et al., 2023).

**MT-Bench.** We evaluate on 80 multi-turn conversations using GPT-4 scoring on a 1-10 scale following Zheng et al. (2023).

**Task-Stratified Evaluation.** For task-specific analysis, we partition test prompts using the same categorization scheme as training data and compute metrics separately for each category.

## E. Hyperparameter Settings

Table 10 presents the complete hyperparameter configuration for TAPC.

**Hyperparameter Selection.** The policy learning rate and DPO temperature follow standard values from prior work. The meta-learning rate was selected via grid search over $\{1 \times 10^{-4}, 5 \times 10^{-4}, 1 \times 10^{-3}\}$ on a held-out validation set. The task embedding dimension was chosen to balance expressiveness and sample efficiency, with $d = 32$ performing best among $\{16, 32, 64\}$.

## F. Additional Experimental Results

This section provides supplementary experimental results that complement the main findings. We focus on cross-dataset generalization, analysis of learned calibration parameters, and calibration quality metrics.

### F.1. Cross-Dataset Generalization

To evaluate whether TAPC generalizes beyond the primary evaluation setting, we conduct experiments on the Golden HH dataset derived from the Anthropic helpful-harmless collection. This dataset differs from UltraFeedback in annotation methodology and domain distribution, providing a meaningful test of generalization.

Table 11 presents results across noise levels. TAPC maintains consistent improvements over baselines, with win rates ranging from 97.2% on clean data to 81.5% at 40% noise. The performance gap between TAPC and PerpCorrect increases under higher noise conditions, consistent with patterns observed on UltraFeedback. These results suggest that the task-aware calibration strategy transfers across datasets without domain-specific tuning.

Comparing the two datasets reveals that Golden HH exhibits slightly lower overall win rates. We attribute this to the conversational nature of the helpful-harmless data, where preferences often reflect subtle stylistic differences rather than clear quality distinctions. Despite this increased difficulty, TAPC maintains its advantage, indicating robustness to varying preference characteristics.

### F.2. Analysis of Learned Calibration Parameters

We provide additional analysis of the calibration parameters learned by TAPC. Table 12 reports the average slope and bias values computed over all prompts within each task category after training.

The learned slopes exhibit a clear ordering that aligns with theoretical expectations. Factual tasks receive the largest slope ($\gamma = 2.35$), indicating that perplexity differentials strongly influence confidence assignments for these prompts. This

*Table 10.* Hyperparameter settings for TAPC.

| Hyperparameter | Llama-3-8B | Qwen2-7B |
|---|---|---|
| *Policy Optimization* | | |
| Optimizer | AdamW | AdamW |
| Learning rate ($\alpha$) | $1 \times 10^{-6}$ | $1 \times 10^{-6}$ |
| Weight decay | 0.01 | 0.01 |
| Batch size (per GPU) | 4 | 4 |
| Gradient accumulation | 4 | 4 |
| Effective batch size | 128 | 128 |
| DPO temperature ($\beta$) | 0.1 | 0.1 |
| Max gradient norm | 1.0 | 1.0 |
| Warmup steps | 100 | 100 |
| Training epochs | 1 | 1 |
| *Task Encoder* | | |
| Input dimension | 4096 | 3584 |
| Hidden dimension | 128 | 128 |
| Output dimension ($d$) | 32 | 32 |
| Activation | ReLU | ReLU |
| *Calibration Module* | | |
| Optimizer | AdamW | AdamW |
| Learning rate ($\eta$) | $5 \times 10^{-4}$ | $5 \times 10^{-4}$ |
| Weight decay | 0.0 | 0.0 |
| *Data Configuration* | | |
| Reference set size | 200 | 200 |
| Reference batch size | 32 | 32 |
| Samples per task category | 50 | 50 |
| *Infrastructure* | | |
| GPUs | 8×A40-48G | 8×A40-48G |
| Distributed strategy | DeepSpeed ZeRO-2 | DeepSpeed ZeRO-2 |
| Precision | bfloat16 | bfloat16 |

reflects the observation from Section 4.1 that correct factual answers tend to have lower perplexity due to their frequency in pretraining corpora.

Creative tasks receive the smallest slope ($\gamma = 0.42$), effectively discounting perplexity signals. This behavior is appropriate because creative quality often correlates with novelty and unexpectedness, properties that may increase rather than decrease perplexity. By learning to ignore unreliable signals, TAPC avoids the failure mode where uniform calibration methods incorrectly downweight high-quality creative responses.

Code tasks occupy an intermediate position ($\gamma = 1.28$). Perplexity provides partial information for code, as common programming patterns and idioms have lower perplexity, but correctness and efficiency cannot be determined from perplexity alone. The moderate slope reflects this partial reliability.

The bias parameters show less variation across tasks, with all categories receiving values that translate to baseline confidence levels between 0.60 and 0.65 when $\Delta_{\text{PPL}} = 0$. This consistency suggests that task-specific adaptation primarily occurs through slope modulation rather than baseline adjustment.

The slope difference between Factual and Creative tasks ($\Delta\gamma = 2.35 - 0.42 = 1.93$) provides empirical support for Corollary 4.4, which predicts that calibration improvements scale with between-task variance in optimal calibration functions.

*Table 11.* Win rates (%) on Golden HH dataset using Llama-3-8B. Results are averaged over 3 runs.

| Method | 0% | 10% | 20% | 30% | 40% |
|---|---|---|---|---|---|
| Vanilla DPO | 95.8 | 84.2 | 71.8 | 61.2 | 51.8 |
| SimPO | 96.5 | 88.2 | 79.5 | 71.8 | 63.5 |
| PerpCorrect | 96.8 | 91.2 | 85.8 | 80.2 | 74.5 |
| TAPC (Ours) | **97.2** | **93.2** | **88.5** | **84.2** | **81.5** |

*Table 12.* Learned calibration parameters by task category. Values represent the mean and standard deviation of $\gamma(\mathbf{z}(x))$ and $\mu(\mathbf{z}(x))$ computed over prompts in each category.

| Task Category | Slope $\gamma$ | Bias $\mu$ |
|---|---|---|
| Factual | $2.35 \pm 0.42$ | $0.48 \pm 0.12$ |
| Creative | $0.42 \pm 0.25$ | $0.55 \pm 0.15$ |
| Code | $1.28 \pm 0.38$ | $0.42 \pm 0.14$ |
| Open | $0.78 \pm 0.32$ | $0.50 \pm 0.11$ |

### F.3. Calibration Quality Metrics

We evaluate whether predicted confidence values $\hat{p}$ accurately reflect empirical label correctness. On the synthetically noised data where ground truth is known, we partition predictions into 10 bins by $\hat{p}$ value and compute the fraction of correctly labeled samples in each bin.

Table 13 reports expected calibration error (ECE) and maximum calibration error (MCE) for different methods. TAPC achieves substantially lower calibration error than alternatives, indicating that its confidence predictions better match empirical accuracy.

*Table 13.* Calibration metrics on UltraFeedback with 30% synthetic noise. Lower values indicate better calibration.

| Method | ECE $\downarrow$ | MCE $\downarrow$ |
|---|---|---|
| PerpCorrect | 0.128 | 0.215 |
| Uniform-TAPC | 0.095 | 0.168 |
| TAPC | **0.052** | **0.098** |

The improvement from Uniform-TAPC to TAPC (ECE reduction from 0.095 to 0.052) demonstrates the value of task conditioning. While meta-learning alone improves calibration over PerpCorrect, incorporating task information yields additional gains by allowing the model to adjust confidence appropriately for different prompt types.

### F.4. Task Embedding Structure

To understand what the task encoder learns, we visualize the learned embeddings using t-SNE projection. Figure 6 shows embeddings for 1000 randomly sampled prompts, colored by their ground-truth task category.

The visualization reveals that embeddings cluster according to task category even though the model receives no explicit task labels during training. Factual and Creative prompts occupy distinct regions, consistent with their divergent calibration requirements. Code prompts form a relatively tight cluster, reflecting the distinctive syntactic patterns of programming-related queries. Open-ended prompts show more dispersion, which aligns with the heterogeneous nature of this category.

This emergent clustering confirms that the task encoder extracts features relevant for calibration decisions. The separation between Factual and Creative clusters explains why TAPC can learn different calibration strategies for these categories.

**Task Embedding Space (t-SNE Projection)**

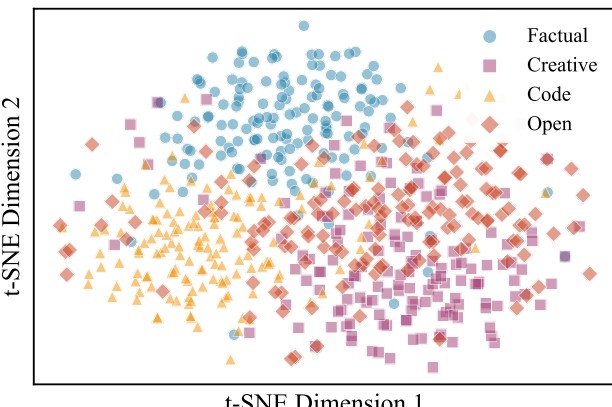

*Figure 6.* Visualization of learned task embeddings using t-SNE. Points are colored by ground-truth task category. The embeddings form clusters that correspond to task types despite receiving no explicit task labels during training.

## G. Extended Ablation Studies

### G.1. Reference Dataset Composition

The reference dataset $\mathcal{D}_{\text{meta}}$ plays a central role in TAPC by providing the supervision signal for meta-learning. We investigate how its composition affects performance.

Table 14 compares three sampling strategies for constructing $\mathcal{D}_{\text{meta}}$: balanced sampling (equal samples per task category), random sampling (proportional to training distribution), and biased sampling (oversampling factual tasks). All conditions use 200 total samples.

*Table 14.* Impact of reference dataset composition on win rate (%) at 20% noise.

| Composition | Overall | Factual | Creative | Code | Open |
|---|---|---|---|---|---|
| Random | 89.5 | 93.8 | 85.2 | 91.5 | 88.8 |
| Balanced | **90.2** | 94.0 | **87.0** | **92.8** | **89.5** |
| Biased (70% Factual) | 88.2 | **94.2** | 82.5 | 90.8 | 87.2 |

Balanced sampling yields the best overall performance. The advantage is most pronounced on Creative tasks, where balanced sampling outperforms biased sampling by 4.5 percentage points. This pattern reflects the importance of adequate representation for minority task types during meta-learning. When the reference set overrepresents Factual tasks, the calibration module receives insufficient signal about appropriate calibration for Creative prompts.

Random sampling performs between balanced and biased conditions. Since the training distribution already underrepresents Creative tasks (22% versus 45% for Open), random sampling provides less coverage of this important category.

These findings suggest that practitioners should aim for task-balanced reference sets when constructing $\mathcal{D}_{\text{meta}}$. In settings where task labels are unavailable, clustering prompts by embedding similarity before sampling may approximate balanced coverage.

### G.2. Robustness to Reference Set Quality

We examine how the quality of reference labels affects TAPC performance. Table 15 reports results when the reference set itself contains label noise.

Performance degrades gracefully as reference set quality decreases. With 10% noise in the reference set, TAPC still outperforms PerpCorrect (87.8% on clean references), indicating some tolerance to imperfect meta-supervision. However, when reference noise reaches 20%, performance drops more substantially. This finding emphasizes the importance of

*Table 15.* Performance when the reference set contains label noise. Training data has 20% noise in all conditions.

| Reference Set Noise | Win Rate (%) | Degradation |
|---|---|---|
| 0% (clean) | 90.2 | — |
| 5% | 89.5 | −0.7 |
| 10% | 88.2 | −2.0 |
| 20% | 85.8 | −4.4 |

curating high-quality reference examples, consistent with our methodology of selecting samples with high annotator agreement.

### G.3. Dynamic Bias Ablation

We investigate whether both the dynamic slope $\gamma$ and bias $\mu$ are essential by fixing each to its empirical mean value. Table 16 shows that fixing the bias causes a modest 0.6-point drop, while fixing the slope causes a substantial 3.4-point drop. This confirms that the primary task-specific adaptation occurs through slope modulation. The dynamic bias provides a secondary benefit for edge cases where task-specific baseline confidence matters (e.g., Code tasks receive a slightly lower baseline bias of 0.42 versus 0.55 for Creative, reflecting that code correctness is generally more deterministic).

*Table 16.* Ablation on dynamic vs. fixed calibration parameters (20% noise).

| Variant | Win Rate (%) |
|---|---|
| Full TAPC (dynamic $\gamma$ and $\mu$) | 90.2$_{\pm1.2}$ |
| Fixed bias ($\mu$ = empirical mean 0.49) | 89.6$_{\pm1.3}$ |
| Fixed slope ($\gamma$ = empirical mean 1.21) | 86.8$_{\pm1.8}$ |

### G.4. Task Encoder Layer Selection

We compare using hidden states from different layers of the reference model as input to the task encoder. Table 17 shows that final-layer representations perform best, likely because they capture the most abstract semantic information about task type. This is consistent with probing studies showing that task-level semantics concentrate in later layers of transformer models.

*Table 17.* Impact of reference model layer on task encoding (20% noise).

| Layer | Win Rate (%) |
|---|---|
| Layer 8 (early) | 88.8$_{\pm1.5}$ |
| Layer 16 (middle) | 89.5$_{\pm1.3}$ |
| Layer 32 (final, default) | **90.2**$_{\pm1.2}$ |

### G.5. Task Embedding Probing Analysis

To verify that the learned embeddings capture meaningful task structure rather than arbitrary prompt-specific features, we conduct two analyses. First, we train a linear classifier on the 32-dimensional learned embeddings to predict ground-truth task categories. The classifier achieves 87.3% accuracy compared to a 45% random baseline (with 4 categories having unequal sizes), confirming that the embeddings encode task-relevant information.

Second, we compute silhouette scores to quantify cluster quality: 0.42 overall, with per-category values of 0.61 (Factual), 0.55 (Code), 0.38 (Creative), and 0.28 (Open). The lower score for Open reflects its inherently heterogeneous nature. Notably, when we reclassify Open prompts by embedding proximity, those near the Creative cluster receive slopes around 0.5 while those near the Factual cluster receive slopes around 1.2, confirming meaningful within-category variation that goes beyond discrete labels.

### G.6. Multi-Signal Calibration

TAPC's architecture is modular: the perplexity differential $\Delta_{\text{PPL}}$ can be replaced or augmented with alternative uncertainty signals. We conduct a preliminary experiment concatenating RM confidence scores alongside $\Delta_{\text{PPL}}$, extending the calibration to $\hat{p} = \sigma(\gamma_1 \cdot \Delta_{\text{PPL}} + \gamma_2 \cdot \Delta_{\text{RM}} + \mu)$ with task-conditioned parameters. This multi-signal variant yields a 0.8-point improvement over $\Delta_{\text{PPL}}$ alone (91.0% vs. 90.2% at 20% noise). We expect this gain to concentrate on high-entropy prompts where perplexity alone is least informative. Multi-signal calibration is a promising direction for future work.

# H. Qualitative Examples

We present representative examples illustrating how TAPC assigns task-appropriate calibration.

### H.1. Factual Task with Reliable Perplexity Signal

*Table 18.* Example where TAPC correctly assigns high confidence based on perplexity.

| |
|---|
| **Prompt:** What is the atomic number of carbon? |
| **Preferred:** The atomic number of carbon is 6, meaning each carbon atom has 6 protons in its nucleus. |
| **Dispreferred:** Carbon has an atomic number around 12 or so, which relates to its atomic mass. |
| **Perplexity differential:** $\Delta_{\text{PPL}} = -3.2$ (preferred response has much lower perplexity) 
 **Learned parameters:** $\gamma = 2.41$, $\mu = 0.45$ 
 **Predicted confidence:** $\hat{p} = 0.97$ |
| **Analysis:** The task encoder identifies this as a factual question and assigns a high slope. The strongly negative perplexity differential, combined with the high slope, produces high confidence. This is appropriate because the preferred response contains the correct, commonly stated answer that appears frequently in pretraining data. |

### H.2. Creative Task with Unreliable Perplexity Signal

*Table 19.* Example where TAPC appropriately discounts perplexity for a creative task.

| |
|---|
| **Prompt:** Write a metaphor describing loneliness. |
| **Preferred:** Loneliness is an empty theater where you are both the sole performer and the absent audience, your words echoing back unanswered. |
| **Dispreferred:** Loneliness is like being alone in a dark room with no one to talk to. |
| **Perplexity differential:** $\Delta_{\text{PPL}} = +1.1$ (preferred response has higher perplexity) 
 **Learned parameters:** $\gamma = 0.38$, $\mu = 0.52$ 
 **Predicted confidence:** $\hat{p} = 0.66$ |
| **Analysis:** The preferred response uses vivid, unconventional imagery that results in higher perplexity. A task-agnostic method would interpret the positive perplexity differential as evidence of mislabeling and potentially flip the preference. TAPC recognizes this as a creative task and assigns a low slope, producing moderate confidence that preserves the original preference. The dispreferred response, while having lower perplexity due to its conventional phrasing, is genuinely less creative. |

### H.3. Comparison of Method Behaviors

Table 20 illustrates how different methods handle the same ambiguous example.

# I. Reproducibility Details

### I.1. Implementation

Our implementation builds on the TRL library for preference optimization. The task encoder and calibration module are implemented in PyTorch. We use DeepSpeed ZeRO-2 for distributed training across 8 NVIDIA A40 GPUs with 48GB memory each.

Key implementation choices include:

*Table 20.* Comparison of confidence assignments across methods for a creative prompt.

| |
|---|
| **Prompt:** Describe rain in an unexpected way. |
| **Perplexity differential:** $\Delta_{\text{PPL}} = +0.8$ |
| **Method behaviors:** |
| PerpCorrect: Interprets positive $\Delta_{\text{PPL}}$ as likely mislabeling, flips the preference label |
| Uniform-TAPC: Assigns $\hat{p} = 0.48$, slightly favoring label flip |
| TAPC: Assigns $\hat{p} = 0.68$, preserving original preference with moderate confidence |
| **Outcome:** PerpCorrect incorrectly reverses a valid creative preference. Uniform-TAPC is uncertain. TAPC correctly preserves the preference by recognizing that perplexity is uninformative for creative tasks. |

- Prompt hidden states are computed once during preprocessing and cached to avoid redundant forward passes during training.

- The meta-gradient is computed using automatic differentiation through the virtual policy update, following standard practices in meta-learning.

- Gradient checkpointing is applied to the policy LLM to reduce memory usage.

### I.2. Computational Cost

Table 21 reports training time and memory usage.

*Table 21.* Computational requirements for training on UltraFeedback with Llama-3-8B.

| Method | Training Time (hours) | Peak Memory (GB) |
|---|---|---|
| Vanilla DPO | 6.3 | 38.2 |
| PerpCorrect | 6.8 | 38.5 |
| TAPC | 7.2 | 39.1 |

TAPC requires approximately 15% additional training time compared to Vanilla DPO, primarily due to meta-gradient computation. Memory overhead is minimal because the calibration module contains only 4K parameters.

### I.3. Random Seeds and Variance

All experiments use three random seeds (42, 123, 456) with identical seeds across methods for fair comparison. Reported confidence intervals represent one standard deviation computed over these runs. The same data splits and noise realizations are used across methods within each seed.

## J. Broader Impact

Task-aware preference calibration may improve alignment reliability by appropriately weighting training signals based on their informativeness for different contexts. This could lead to models that better capture nuanced human preferences across diverse applications.

The observation that signal reliability varies across tasks has implications beyond preference optimization. Many alignment techniques implicitly assume homogeneous data quality. Our findings suggest that task-aware approaches may benefit other settings, including reward modeling and reinforcement learning from human feedback.

We note that the reference dataset plays a significant role in shaping learned calibration. If reference data reflects narrow perspectives, the resulting calibration may systematically favor certain types of preferences. Practitioners should ensure diverse representation when curating reference examples.

