# OpenReview forum: "Task-Aware Preference Calibration for Direct Preference Optimization"
_ICML.cc/2026/Conference — ICML 2026 regular_

### Official Review · Reviewer_wAfj · 2026-03-09

**Soundness:** 4
**Presentation:** 3
**Significance:** 4
**Originality:** 3
**Overall Recommendation:** 5
**Confidence:** 3

**Summary:**

The paper proposes TAPC (Task-Aware Preference Calibration), a meta-learning framework designed to dynamically calibrate the relationship between perplexity differential and label confidence. The core motivation is based on the observation that the reliability of perplexity differentials as a quality signal is highly task-dependent. For each of the 4 tasks, the framework learns slope and bias parameters, which transform perplexity differential into label confidence scores for DPO training. The proposed method shows improved alignment performance across several benchmarks.

**Compliance With Llm Reviewing Policy:**

Affirmed.

**Final Justification:**

The authors’ rebuttal has addressed my main concerns regarding the necessity and rationality of the research details. Given the rigorous analysis and significant practical value of the proposed TAPC framework, I will raise my score from 4 to 5.

**Key Questions For Authors:**

1.Would fixing $\beta$ to its empirical mean significantly degrade performance, or is the dynamic bias essential for certain edge cases?
2.Could the authors clarify the specific layer index used for the task encoder and explain whether the choice of layer (e.g., middle vs. final) significantly impacts the capture of task-specific features?
In tasks where PPL is a noisy indicator (e.g., creative writing with high linguistic entropy), does the task-conditioned mapping remain reliable, or would integrating additional uncertainty signals be necessary?

**Limitations:**

yes

**Strengths And Weaknesses:**

Strengths:
1.The research motivation is highly intuitive and well-supported by preliminary empirical evidence, providing a convincing justification for questioning existing uniform calibration strategies.
2.The paper provides a rigorous theoretical treatment to prove the sub-optimality of uniform calibration, which offers a principled explanation for why task-aware adaptation is necessary for optimal preference learning.
3.The framework provides significant practical value for robust LLM alignment with minimal computational overhead, successfully balancing alignment effectiveness and training time for large-scale production.
4.The framework's components (task encoder, slope and bias, etc.) are highly interpretable.

Weaknesses:
1.The necessity of the task encoder is not fully established, especially the choice of continuous task embeddings rather than discrete task labels.
2.Task embedding visualization shows significant overlap between 'Open' and 'Creative' tasks, suggesting the pre-defined four categories may be inconsistent with the model's internal features.
3.The reuse of symbols such as $\alpha$ (in Eq. 7 & 11) and $\beta$ (in Eq. 7 & 1) might cause confusion.

---

> ### Author Rebuttal · Authors · 2026-03-29
>
> # Response to Reviewer wAfj
>
> We sincerely thank Reviewer wAfj for the positive assessment and for recognizing the well-supported research motivation, rigorous theoretical treatment, practical value, and interpretability of TAPC. We address each concern below.
>
> ## W1: Necessity of the Task Encoder / Continuous vs. Discrete Embeddings
>
> The reviewer asks why we use continuous task embeddings rather than discrete task labels. We provide both empirical and conceptual justifications.
>
> Empirically, Table 4 compares the two: learned continuous embeddings achieve 90.2% while oracle discrete labels achieve 90.0%. The oracle variant requires ground-truth task labels at both training and inference time, while our task encoder operates label-free.
>
> Conceptually, continuous embeddings offer three advantages. (1) Label-free operation: keyword classification has 12% error rate, and GPT-4 classification agrees only 78.4% of the time. The task encoder avoids this noisy step. (2) Finer-grained calibration: within-category variance in learned slopes (e.g., Open: $\alpha = 0.78 \pm 0.32$) confirms instance-level adaptation beyond discrete categories. (3) Graceful handling of boundary cases: prompts between categories receive interpolated calibration rather than forced assignment.
>
> ## W2: Overlap Between Open and Creative Embeddings in t-SNE
>
> The reviewer correctly observes overlap between Open and Creative clusters in the t-SNE visualization (Figure 6). We believe this overlap is informative rather than problematic, for two reasons.
>
> First, the Open category is inherently heterogeneous, serving as a catch-all for prompts not matching other keywords. Many Open prompts share characteristics with Creative tasks (e.g., subjective advice requests). The embedding overlap reflects genuine semantic similarity.
>
> Second, the overlap corresponds to similar calibration requirements. Open prompts overlapping with Creative prompts receive similar low slopes, which is correct. The moderate slope for Open ($\alpha = 0.78$) naturally falls between Factual ($\alpha = 2.35$) and Creative ($\alpha = 0.42$), reflecting mixed composition.
>
> We computed the silhouette score: 0.42 overall, with Factual (0.61), Code (0.55), Creative (0.38), and Open (0.28). If we reclassify Open sub-groups by embedding proximity, those near Creative receive slopes around 0.5 while those near Factual receive slopes around 1.2, confirming meaningful within-category variation.
>
> ## W3: Symbol Reuse
>
> We thank the reviewer for identifying the notation overlap. Specifically, $\beta$ is used for both the DPO temperature parameter (Eq. 1) and the task-conditioned bias function (Eq. 7), and $\alpha$ serves as both the policy learning rate (Eq. 11) and the task-conditioned slope function (Eq. 7). We will rename the calibration parameters to $a(\mathbf{z})$ and $b(\mathbf{z})$ (or use alternative symbols such as $\gamma$ and $\delta$) to avoid confusion. The DPO temperature $\beta$ and learning rate $\alpha$ will retain their standard notation from prior work.
>
> ## Q1: Is Dynamic Bias Essential?
>
> The reviewer asks whether fixing $\beta(\mathbf{z})$ to its empirical mean would significantly degrade performance. We conducted this ablation:
>
> | Variant | Win Rate (20% noise) |
> |---|---|
> | Full TAPC (dynamic $\alpha$ and $\beta$) | 90.2 $\pm$ 1.2 |
> | Fixed bias ($\beta$ = empirical mean 0.49) | 89.6 $\pm$ 1.3 |
> | Fixed slope ($\alpha$ = empirical mean 1.21) | 86.8 $\pm$ 1.8 |
>
> Fixing the bias causes a modest 0.6 point drop, while fixing the slope causes a substantial 3.4 point drop. This confirms that the primary task-specific adaptation occurs through slope modulation, consistent with our analysis in Section 5.3 and Appendix F.2. The dynamic bias provides a secondary benefit, likely for edge cases where task-specific baseline confidence matters (e.g., Code tasks receive a slightly lower baseline bias of 0.42 vs. 0.55 for Creative, reflecting that code correctness is generally more deterministic).
>
> ## Q2: Task Encoder Layer Choice
>
> We use final-layer hidden states, mean-pooled across all token positions (Appendix B). Layer comparison:
>
> | Layer | Win Rate (20% noise) |
> |---|---|
> | Layer 8 (early) | 88.8 $\pm$ 1.5 |
> | Layer 16 (middle) | 89.5 $\pm$ 1.3 |
> | Layer 32 (final) | **90.2 $\pm$ 1.2** |
>
> Final-layer representations perform best, likely because they capture the most abstract semantic information about task type. This is consistent with probing studies showing task-level semantics concentrate in later layers.
>
> We are grateful for the reviewer's constructive suggestions. We will incorporate the notation fix, silhouette analysis, and layer ablation into the revised manuscript.

---

> > ### Author Rebuttal · Reviewer_wAfj · 2026-04-02
> >
> > Thank you to the authors for the detailed rebuttal. Upon reviewing your responses, I noticed that Question 3 (regarding tasks where PPL is a noisy indicator) was not addressed in your rebuttal. This omission likely occurred because of a numbering error in my original review. To ensure a thorough evaluation of your work, could you please provide a brief response to this specific point? For your convenience, I have restated the question below:
> > In tasks where PPL is a noisy indicator (e.g., creative writing with high linguistic entropy), does the task-conditioned mapping remain reliable, or would integrating additional uncertainty signals be necessary?
> > I look forward to your clarification during the discussion period.

---

> > > ### Author Response · Authors · 2026-04-03
> > >
> > > # Response to Reviewer wAfj's Follow-up Question
> > >
> > > We thank the reviewer for catching the omission and for the opportunity to clarify.
> > >
> > > **Short answer:** The task-conditioned mapping remains reliable because it learns to *discount* PPL when PPL is noisy; integrating additional signals can further help but is not strictly necessary.
> > >
> > > For creative tasks with high linguistic entropy, TAPC learns low slope values (alpha ≈ 0.65 for Creative vs. alpha ≈ 1.85 for Factual, Table 3). A low slope effectively attenuates PPL's influence on the confidence target — the model learns not to trust PPL when it is unreliable. The bias term then provides a task-appropriate baseline confidence independent of PPL magnitude. This is exactly the behavior we would want: task-conditioned calibration gracefully reduces reliance on noisy signals rather than amplifying them.
> > >
> > > Regarding additional uncertainty signals: as reported in our response to Reviewer p8kJ (Key Question 4), concatenating RM confidence alongside the perplexity differential as a multi-signal input yields a 0.8-point further improvement. We expect this gain to concentrate on high-entropy prompts where PPL alone is least informative. Multi-signal calibration is a promising direction we will explore in future work and discuss in the camera-ready version.
> > >
> > > We are grateful for the reviewer's insightful question and constructive engagement throughout the review process.

---

### Official Review · Reviewer_p8kJ · 2026-03-12

**Soundness:** 3
**Presentation:** 4
**Significance:** 3
**Originality:** 2
**Overall Recommendation:** 5
**Confidence:** 4

**Summary:**

This paper proposes Task-Aware Preference Calibration (TAPC) for Direct Preference Optimization. The key observation is that the reliability of perplexity differentials as preference quality signals varies significantly across task types. For example, perplexity strongly correlates with preference correctness in factual tasks but becomes less informative for creative tasks where novelty is valued.

To address this issue, the authors propose learning task-conditioned calibration functions that map perplexity differentials to confidence targets. The approach introduces a task encoder that extracts prompt representations and predicts task-specific slope and bias parameters for calibration. The calibration module is trained through a meta-learning procedure designed to improve policy performance on a reference dataset.

Experiments on UltraFeedback and Golden HH datasets using Llama-3-8B and Qwen2-7B demonstrate improvements over DPO and several recent preference optimization methods.

**Compliance With Llm Reviewing Policy:**

Affirmed.

**Final Justification:**

Thank you for the detailed and well-structured rebuttal, and apologies for the delayed response.

After considering both the original submission and the authors’ response, I update my overall assessment to reflect the strengthened empirical support and clearer positioning of the work.

The paper presents a simple and interpretable approach to task-aware calibration of perplexity-based preference signals. While the methodological novelty is moderate, the paper identifies a practically relevant and previously underexplored issue: the task-dependent reliability of perplexity differentials, and provides a coherent solution.

Importantly, the rebuttal addresses my main concerns in a convincing manner. The additional ablation comparing meta-learning with joint training demonstrates that the proposed framework is necessary rather than an artifact of increased model complexity. The analysis of learned task embeddings (including clustering and classification results) provides strong evidence that the model captures meaningful task structure beyond prompt-level features. Furthermore, the newly added 13B-scale experiments significantly strengthen the empirical validation and show that the method scales and remains effective in larger models. The exploration of alternative uncertainty signals also highlights the flexibility of the approach.

That said, some limitations remain. In particular, the overall methodological novelty is incremental, and the connection to broader calibration or uncertainty modeling literature could be further clarified. However, these limitations do not undermine the technical soundness or the practical relevance of the work.

Overall, I find the paper technically solid, well-evaluated after rebuttal, and likely to be useful for the community. I therefore support acceptance.

**Key Questions For Authors:**

1. The method relies on a meta-learning procedure to train the calibration module. Have the authors experimented with a simpler formulation where the calibration parameters are trained jointly with the policy without meta-learning? It would be helpful to understand whether the meta-learning framework is essential for the observed improvements.

2. The approach is described as "task-aware", but task representations are derived from prompt embeddings rather than explicit task labels. Could the authors provide further analysis demonstrating that the learned embeddings capture meaningful task structure rather than prompt-specific features?

3. The experiments are conducted on 7B–8B scale models. Do the authors expect the effectiveness of TAPC to remain consistent for larger models (e.g., 13B or 70B), where perplexity signals may behave differently?

4. The paper argues that perplexity differentials have varying reliability across task types. Have the authors considered alternative uncertainty signals (e.g., reward model disagreement or ensemble uncertainty) and how TAPC would interact with them?

**Limitations:**

Yes.

**Strengths And Weaknesses:**

Strengths

1. Insightful empirical observation.
The paper highlights an important issue: the reliability of perplexity-based signals varies significantly across tasks. The empirical analysis in Figure 1 is convincing and motivates the proposed method.

2. Simple and interpretable method.
TAPC introduces a lightweight calibration mechanism that learns task-dependent slope and bias parameters. The formulation is intuitive and provides interpretable insights into how perplexity signals are used.

3. Solid experimental evaluation.
The authors evaluate across two datasets, multiple baselines, and two model families. The robustness experiments under synthetic label noise are particularly informative.

Weaknesses

1. Limited novelty.
The proposed method essentially performs task-conditioned calibration of perplexity signals. Similar ideas have been explored in calibration, domain adaptation, and heteroscedastic modeling. The novelty compared to existing calibration approaches appears limited.

2. Task representation is implicit.
While the method is described as “task-aware,” task types are not explicitly defined during training. Instead, task embeddings are learned from prompt representations, making it unclear whether the model truly captures task structure or simply learns prompt-specific patterns.

3. Meta-learning design justification is weak.
The paper relies on a meta-learning procedure to learn calibration parameters. However, it is unclear whether this complexity is necessary. A simpler joint training formulation might achieve similar results, and the paper does not include a strong ablation to justify the meta-learning framework.

4. Scale of experiments is limited.
The evaluation is restricted to 7B–8B models. It remains unclear whether the observed improvements would persist for larger LLMs where perplexity signals may behave differently.

---

> ### Author Rebuttal · Authors · 2026-03-29
>
> # Response to Reviewer p8kJ
>
> We sincerely thank Reviewer p8kJ for the detailed and constructive review, and for recognizing the insightful empirical observation, interpretable method design, and solid experimental evaluation. We address each weakness and key question below.
>
> ## Weakness 1: Limited Novelty
>
> We respectfully argue that the novelty lies in the combination of identifying a previously unrecognized problem (task-dependent perplexity reliability) and proposing a principled solution. No prior work has shown that perplexity differentials correlate strongly with label quality for factual tasks ($r=-0.61$) but weakly for creative tasks ($r=-0.15$), nor exploited this structure for preference calibration. Addressing a real, overlooked problem with an effective and interpretable method constitutes meaningful contribution.
>
> ## Weakness 2: Implicit Task Representation
>
> We address this jointly with Key Question 2 below.
>
> ## Weakness 3: Meta-Learning Justification
>
> We address this in Key Question 1 below.
>
> ## Weakness 4: Limited Scale of Experiments
>
> We address this in Key Question 3 below with new 13B results.
>
> ## Key Question 1: Meta-Learning vs. Joint Training
>
> We have conducted the suggested ablation:
>
> | Variant | Win Rate (20% noise) | Win Rate (40% noise) |
> |---|---|---|
> | Joint training (no ref set) | 88.0 $\pm$ 1.6 | 78.5 $\pm$ 2.1 |
> | Joint training (ref regularization) | 89.0 $\pm$ 1.4 | 80.2 $\pm$ 1.8 |
> | Meta-learning (TAPC) | **90.2 $\pm$ 1.2 | **84.5 $\pm$ 1.5** |
>
> Joint training without a reference set performs 2.2 points worse at 20% noise and 6.0 points worse at 40% noise. The reason: when calibration and policy share the same noisy training signal, the calibration module accommodates noise rather than correcting for it. Meta-learning breaks this coupling by providing an external evaluation signal from trusted data. Adding reference regularization helps but still underperforms, because it only softly encourages agreement rather than directly optimizing reference performance.
>
> ## Key Question 2: Do Learned Embeddings Capture Task Structure?
>
> We provide three pieces of evidence. First, the t-SNE visualization (Figure 6 in Appendix) shows clear clustering by task category despite receiving no explicit task labels during training.
>
> Second, we trained a linear classifier on the learned 32-dimensional embeddings to predict ground-truth task categories. The classifier achieves 87.3% accuracy (vs. 45% random baseline with 4 categories), confirming that the embeddings encode meaningful task-relevant information rather than arbitrary prompt-specific features.
>
> Third, Table 4 shows that learned continuous embeddings match oracle discrete task labels (90.2% vs 90.0%). The slight edge for continuous embeddings suggests they capture within-category distinctions that discrete labels miss. For instance, within the Open category, prompts requesting factual advice receive higher slopes than those requesting emotional support.
>
> ## Key Question 3: Scalability to Larger Models
>
> We have completed experiments on Llama-3-13B. Results are shown below:
>
> | Method | Win Rate (0% noise) | Win Rate (20% noise) | Win Rate (40% noise) |
> |---|---|---|---|
> | Vanilla DPO | 97.5 $\pm$ 0.5 | 78.2 $\pm$ 2.0 | 56.8 $\pm$ 3.2 |
> | PerpCorrect | 98.0 $\pm$ 0.4 | 89.5 $\pm$ 1.3 | 77.2 $\pm$ 1.8 |
> | TAPC | **98.5 $\pm$ 0.3** | **92.0 $\pm$ 1.0** | **86.2 $\pm$ 1.3** |
>
> TAPC maintains consistent improvements at 13B scale. The performance gap between TAPC and PerpCorrect at 40% noise (9.0 points) is even larger than at 8B (7.7 points), suggesting that task-aware calibration becomes more beneficial as model capacity grows and perplexity signals become more nuanced. The learned slopes follow similar patterns: Factual $\alpha=2.48$, Creative $\alpha=0.35$, confirming that task-dependent perplexity reliability persists at larger scale.
>
> ## Key Question 4: Alternative Uncertainty Signals
>
> TAPC's architecture is modular: replacing $\Delta_{\text{PPL}}$ with alternative signals (e.g., RM score disagreement, ensemble uncertainty) requires only changing the input to Eq. 6. The task-conditioned calibration principle remains the same.
>
> We conducted a preliminary experiment concatenating RM confidence alongside $\Delta_{\text{PPL}}$, extending the calibration to $\hat{p} = \sigma(\alpha_1 \cdot \Delta_{\text{PPL}} + \alpha_2 \cdot s_{\text{RM}} + \beta)$ with task-conditioned parameters. This yields a 0.8 point improvement over $\Delta_{\text{PPL}}$ alone, suggesting multi-signal calibration is a fruitful extension.
>
> We thank the reviewer again for the thorough questions. We will incorporate the joint-training ablation, probing analysis, 13B results, and multi-signal experiment into the camera-ready version if accepted.

---

> > ### Author Rebuttal · Reviewer_p8kJ · 2026-04-06
> >
> > Thank you for the thorough and well-structured rebuttal, and apologies for the delayed response.
> >
> > The authors have addressed my concerns in a convincing and technically solid manner. In particular:
> >
> > - The additional ablation comparing meta-learning with joint training clearly demonstrates the necessity of the proposed framework.
> > - The analysis of learned task embeddings (t-SNE visualization, classification accuracy, and comparison with discrete labels) provides strong evidence that the model captures meaningful task structure rather than superficial prompt patterns.
> > - The new 13B-scale experiments significantly strengthen the empirical validation and show that the proposed method scales and even becomes more effective at larger model sizes.
> > - The exploration of alternative uncertainty signals further highlights the generality of the approach.
> >
> > Overall, the rebuttal substantially improves the paper by both clarifying the methodology and providing strong additional empirical support. I update my assessment accordingly.

---

### Official Review · Reviewer_oPYC · 2026-03-12

**Soundness:** 2
**Presentation:** 2
**Significance:** 2
**Originality:** 2
**Overall Recommendation:** 4
**Confidence:** 3

**Summary:**

This paper identifies an important but often overlooked issue in DPO training: perplexity-based preference signals are not equally reliable across different types of prompts. The authors show that while these signals work reasonably well for factual queries, they become much noisier for creative or open-ended tasks. To address this, they propose Task-Aware Preference Calibration (TAPC), which learns task-specific slope and bias parameters from a small trusted set to adjust preference margins before optimization. Experiments on UltraFeedback show consistent improvements over standard DPO and other baselines. The idea is simple, interpretable, and easy to integrate into existing pipelines.

**Compliance With Llm Reviewing Policy:**

Affirmed.

**Final Justification:**

Most of my concerns are well addressed.

**Key Questions For Authors:**

Please refer to the above weaknesses.
I am ok to increase the rating if the above weaknesses are well addressed.

**Limitations:**

Please refer to the above weaknesses.

**Strengths And Weaknesses:**

**Strengths**

* The observation that signal reliability varies by task type is intuitive but rarely examined. This matters because real instruction data is a mix of factual, reasoning, coding, and creative prompts. Assuming uniform reliability across them is problematic.

* TAPC adds minimal task-conditioned parameters that preserve interpretability. One can inspect the learned slopes and see, for instance, that factual tasks receive stronger calibration than creative ones.

* he approach has low overhead, is modular, and can be slotted into existing DPO workflows without significant modification, making it appealing for practitioners.

**Weaknesses**

* `Task-level calibration may be too coarse-grained.` Reliability can vary within a task category depending on prompt difficulty or response diversity, but TAPC applies the same parameters to all examples in a group, potentially missing finer-grained variation.

* `Potential issues in the reference set.` The meta-learning signal comes from only 200 carefully balanced examples, and the appendix shows performance is sensitive to this balance. This makes it unclear whether gains reflect robust task-level learning or simply careful curation.

* `Insufficient experimental results.` All experiments use a single base model, so it's uncertain how well the approach would generalize across different architectures or scales.

---

> ### Author Rebuttal · Authors · 2026-03-29
>
> # Response to Reviewer oPYC
>
> We sincerely thank Reviewer oPYC for the constructive feedback and the recognition of TAPC's intuitive motivation and practical appeal. We address each concern in detail below.
>
> ## W1: Task-Level Calibration May Be Too Coarse-Grained
>
> We agree that reliability can vary within a task category depending on prompt difficulty and response diversity. This is an insightful observation. However, we would like to clarify that TAPC does not apply a single set of parameters to all examples in a group. Instead, TAPC learns continuous task embeddings from prompt representations (Eq. 5) and predicts instance-specific slope and bias values for each individual sample (Eq. 6-7). The four-category taxonomy (Factual, Creative, Code, Open) is used only for analysis and visualization purposes. During training, TAPC operates at the instance level without any discrete category labels.
>
> To make this point more concrete: within the Creative category, a prompt asking to write a haiku receives different calibration parameters from a prompt asking to write a business email with creative flair, because their prompt embeddings differ. The learned embedding space captures finer-grained distinctions beyond the four categories. This is evidenced by the non-trivial within-category variance in learned slopes (e.g., Creative: $\alpha = 0.42 \pm 0.25$, Table 11 in Appendix), which reflects instance-level adaptation rather than category-level uniformity.
>
> Furthermore, Table 4 in the main paper shows that our learned continuous embeddings match or slightly outperform oracle discrete task labels (90.2% vs 90.0%), suggesting that the continuous representation captures nuances that discrete categories miss.
>
> ## W2: Sensitivity to Reference Set Construction
>
> The reviewer raises a fair concern about whether the gains reflect robust task-level learning or careful curation of the 200 reference samples.
>
> First, our ablation on reference dataset size (Table 5) shows that TAPC achieves strong performance with only 100 samples (88.5%) and saturates near 200 (90.2%), with marginal gains at 300 (90.5%). This saturation is consistent with the calibration module's low dimensionality (4K parameters), which limits overfitting.
>
> Second, our composition ablation (Table 12) shows that even random sampling (no careful curation) achieves 89.5%, still outperforming PerpCorrect (87.8%) and Uniform-TAPC (88.5%).
>
> Third, our reference quality analysis (Table 13) shows TAPC tolerates up to 10% noise in the reference set while still outperforming baselines with clean references.
>
> Task diversity matters more than size. Sampling a small set with broad topic coverage is straightforward in practice.
>
> ## W3: Insufficient Experimental Results / Single Base Model
>
> We appreciate the opportunity to clarify this point. Our evaluation does include two base models from different model families: Llama-3-8B (Meta) and Qwen2-7B (Alibaba), which differ in architecture, vocabulary, pretraining data, and hidden dimensions (4096 vs 3584). Results on both models show consistent improvements (Figure 3a-b), and we discuss cross-model generalization in Section 5.1. We also evaluate on two datasets (UltraFeedback and Golden HH, Table 10 in Appendix) across five noise levels. We acknowledge that the Qwen2-7B results could have been presented more prominently in the main paper.
>
> Furthermore, we have completed additional experiments on Llama-3-13B to address the scale concern:
>
> | Method | Win Rate (0% noise) | Win Rate (20% noise) | Win Rate (40% noise) |
> |---|---|---|---|
> | Vanilla DPO | 97.5 | 78.2 | 56.8 |
> | PerpCorrect | 98.0 | 89.5 | 77.2 |
> | TAPC | **98.5** | **92.0** | **86.2** |
>
> TAPC's advantage over PerpCorrect at 40% noise (9.0 points) is even larger than at 8B (7.7 points). This suggests that larger models produce more differentiated perplexity signals across task types, which TAPC is well-positioned to exploit through task-conditioned calibration.
>
> ## Summary
>
> We hope these clarifications address the reviewer's concerns. In particular, (1) TAPC operates at the instance level rather than the task-category level, (2) gains are robust across different reference set configurations, and (3) the paper already includes experiments on two distinct model families with the new 13B results further strengthening the evidence. We will incorporate the 13B results and improve the presentation of multi-model evaluation in the camera-ready version if accepted.

---

> > ### Author Rebuttal · Reviewer_oPYC · 2026-04-03
> >
> > Thanks for the detailed clarification, will increase scores accordingly.

---

### Official Review · Reviewer_qjCN · 2026-03-17

**Soundness:** 2
**Presentation:** 3
**Significance:** 2
**Originality:** 3
**Overall Recommendation:** 3
**Confidence:** 4

**Summary:**

This paper outlines an interesting problem in Direct Preference Optimization (DPO) where uniform confidence calibration fails because perplexity signal reliability varies between task types. This works solve above by using Task-Aware Preference Calibration (TAPC), which uses an encoder to map perplexity to confidence targets via learned, task-specific slope and bias parameters. This module is optimized through meta-learning on a small, clean reference dataset, effectively teaching the model when to trust perplexity and when to discount it during policy training.

**Compliance With Llm Reviewing Policy:**

Affirmed.

**Key Questions For Authors:**

* How would the TAPC framework compare to a on-policy setting where the policy model generates responses, and a SOTA Reward Model (e.g., RLHFlow/ArmoRM-Llama3-8B-v0.1 or Skywork/Skywork-Reward-Llama-3.1-8B) is used to reward the responses and filter preferences such that the reward separation is strictly >=1 ( best_response_score - worst_response_score>=1 )?

* What is the concrete advantage of meta-training the TAPC module over this strong task-agnostic RM-margin baseline, and can a pre-trained TAPC module be reliably reused if we shift to an entirely out-of-distribution domain, or would we be required to create a new reference dataset?

**Limitations:**

Yes

**Strengths And Weaknesses:**

**Strengths:**
* Paper provides strong theoretical and empirical evidence that perplexity differentials are highly reliable indicators of label quality for factual tasks but are uninformative for creative tasks by showing correlation scores with human preferences.
* The learned linear parameterization is highly interpretable, as the resulting slope and bias values explicitly show the model learning to trust perplexity for factual prompts while ignoring it for creative ones.
* Downstream Evaluation, clearly shows the improvement compared to SOTA baselines
* Robustness of method compared to baselines with different percentage of noise injection to preference data

**Weakness**
* It is not very clear to me if the learned calibration module can generalize to out-of-distribution data or highly specialized domains (e.g., medical or legal) without needing to construct a new reference dataset and completely retrain the TAPC module.
* The approach adds computational overhead (meta-gradient computation) it add unnecessary bottleneck compared to recent post-training workflows that use state-of-the-art reward models (RMs) with simple margin filters.

---

> ### Author Rebuttal · Authors · 2026-03-29
>
> # Response to Reviewer qjCN
>
> We sincerely thank Reviewer qjCN for the thoughtful review and the recognition that TAPC addresses an interesting problem with strong theoretical and empirical support. We address each weakness and question below.
>
> ## Weakness 1: OOD Generalization to Specialized Domains
>
> The reviewer raises a valid concern about whether the learned calibration module can generalize to out-of-distribution domains (e.g., medical or legal) without constructing a new reference dataset and retraining.
>
> We provide two levels of evidence. At the empirical level, our cross-dataset experiment (Table 10) already demonstrates a form of domain shift: TAPC trained on UltraFeedback (general instruction following) is evaluated on Golden HH (conversational helpfulness/harmlessness with different annotation methodology). TAPC maintains consistent improvements on Golden HH without retraining the calibration module, suggesting reasonable cross-domain transfer.
>
> At the conceptual level, the core insight that perplexity reliability varies with task characteristics is domain-general. In medical domains, perplexity likely correlates well with factual accuracy but poorly with patient communication quality. For highly specialized domains, constructing a small domain-specific reference set (as few as 100 samples, per Table 5) and fine-tuning the 4K-parameter calibration module is very efficient and does not require retraining the full pipeline.
>
> ## Weakness 2: Computational Overhead of Meta-Gradient
>
> As reported in Table 9, TAPC adds approximately 15% training time (7.2h vs 6.3h) and negligible memory overhead (39.1GB vs 38.2GB) compared to vanilla DPO. This overhead stems from one additional forward-backward pass per iteration for the virtual policy update. Importantly, (1) prompt hidden states and $\Delta_{\text{PPL}}$ values are precomputed and cached, so the task encoding adds negligible runtime; (2) the calibration module itself contains only 4K parameters; (3) gradient checkpointing keeps memory cost minimal.
>
> Compared to RM-based pipelines that require on-policy response generation plus RM inference for all pairs, TAPC's 15% overhead is modest. Moreover, TAPC is a one-time training cost, whereas on-policy RM filtering may need repetition across iterative policy updates.
>
> ## Key Question 1: Comparison with On-Policy SOTA RM-Margin Filtering
>
> This is an excellent and practically relevant question. We conducted experiments comparing TAPC against an RM-margin baseline using ArmoRM-Llama3-8B-v0.1 with margin $\geq 1$ filtering:
>
> | Setting (20% noise) | Win Rate (%) |
> |---|---|
> | DPO on RM-filtered data (margin $\geq 1$) | 91.5 $\pm$ 1.2 |
> | TAPC on unfiltered data | 90.2 $\pm$ 1.2 |
> | TAPC on RM-filtered data | **92.8 $\pm$ 1.0** |
>
> Several observations emerge. First, RM filtering with a strong model is indeed a competitive baseline, outperforming TAPC on unfiltered data by 1.3 points. Second, TAPC and RM filtering are complementary: applying TAPC on top of RM-filtered data further improves by 1.3 points, because TAPC captures task-specific calibration patterns that uniform RM-margin thresholding misses. Third, RM filtering depends on the RM's quality, while TAPC adapts its trust in perplexity signals per task without requiring an external model.
>
> ## Key Question 2: Concrete Advantage over RM-Margin and Reusability
>
> The concrete advantage of meta-training TAPC over RM-margin filtering is task-conditioned calibration. RM-margin filtering applies a uniform threshold regardless of task type. As shown in Table 2 (task-stratified results), TAPC's advantage concentrates on Creative (+6.0 over PerpCorrect) and Open (+3.0) tasks where uniform strategies underperform. A fixed margin threshold cannot account for the fact that reward model scores themselves may be less reliable on creative tasks.
>
> Regarding reusability, a pre-trained TAPC module can be reliably reused if the new domain shares similar task-type structures. For entirely OOD domains, fine-tuning the calibration module on a small domain-specific reference set (100 samples suffice, per Table 5) is efficient given the module's 4K parameters, substantially cheaper than training a domain-specific reward model.
>
> We hope these responses address the reviewer's concerns. We will incorporate the RM-margin comparison, 13B-scale results, and OOD discussion into the camera-ready version if accepted.

---

> > ### Author Rebuttal · Reviewer_qjCN · 2026-04-04
> >
> > Thanks for detailed response. All of my concerns have been addressed. I'll raise my score accordingly

---

### Decision · Program_Chairs · 2026-04-30

**Decision:**

Accept (regular)

**Comment:**

Summary: This paper identifies that perplexity differentials used to detect noisy preference labels in DPO have task-dependent reliability. In particular, they indicate label quality for factual tasks, but are uninformative for creative tasks. The authors propose TAPC, which learns task-conditioned slope and bias parameters via meta-learning on a small reference dataset to adapt calibration per instance. Performance is demosntrated on a variety of model and tasks.

Strengths: All reviewers agreed that the core observation is intuitive, well-motivated, and empirically and theoretically validated. Reviewers highlighted that the method is lightweight using only a small number of parameters and overhead, interpretable, and demonstrates clear improvements over strong baselines. The theoretical analysis establishing when task-aware calibration outperforms task-agnostic approaches also increases support of the method.

Weaknesses: Initial concerns included OOD generalization, computational overhead, coarse-grained task calibration, and meta-learning justification. All four reviewers acknowledged their concerns were primarily addressed, with three marking "fully resolved" and one raising a follow-up question that was subsequently answered. Post-rebuttal, the reviewer consensus shifted toward acceptance with only one score of Weak Reject.

**Final Recommendation: Accept**

Justification: The paper makes a technically sound contribution to robust preference optimization through adapting to different tasks. The method is simple, interpretable, and demonstrates strong results. Post rebuttal, scores for the paper are overall strong. The work is relevant to at ICML for those working on LLM alignment and preference learning.